# Toward a more informative representation of the fetal–neonatal brain connectome using variational autoencoder

Jung-Hoon Kim*, Josepheen De Asis-Cruz, Dhineshvikram Krishnamurthy, Catherine Limperopoulos*

Developing Brain Institute, Children's National Hospital, Washington, United States

**Abstract** Recent advances in functional magnetic resonance imaging (fMRI) have helped elucidate previously inaccessible trajectories of early-life prenatal and neonatal brain development. To date, the interpretation of fetal–neonatal fMRI data has relied on linear analytic models, akin to adult neuroimaging data. However, unlike the adult brain, the fetal and newborn brain develops extraordinarily rapidly, far outpacing any other brain development period across the life span. Consequently, conventional linear computational models may not adequately capture these accelerated and complex neurodevelopmental trajectories during this critical period of brain development along the prenatal-neonatal continuum. To obtain a nuanced understanding of fetal–neonatal brain development, including nonlinear growth, for the first time, we developed quantitative, systems-wide representations of brain activity in a large sample (>500) of fetuses, preterm, and full-term neonates using an unsupervised deep generative model called variational autoencoder (VAE), a model previously shown to be superior to linear models in representing complex resting-state data in healthy adults. Here, we demonstrated that nonlinear brain features, that is, latent variables, derived with the VAE pretrained on rsfMRI of human adults, carried important individual neural signatures, leading to improved representation of prenatal-neonatal brain maturational patterns and more accurate and stable age prediction in the neonate cohort compared to linear models. Using the VAE decoder, we also revealed distinct functional brain networks spanning the sensory and default mode networks. Using the VAE, we are able to reliably capture and quantify complex, nonlinear fetal–neonatal functional neural connectivity. This will lay the critical foundation for detailed mapping of healthy and aberrant functional brain signatures that have their origins in fetal life.

*For correspondence:
jkim9@childrensnational.org
(J-HK);
climpero@childrensnational.org
(CL)

Competing interest: The authors declare that no competing interests exist.

## Editor's evaluation

Presenting important findings, this study describes the development of the functional brain connectome in human fetuses and neonates through the application of a novel deep learning approach: adult trained variational autoencoder. The methodology, analyses, and evidence provided are convincing and pave the way for future studies on non-linear models of brain network maturation. This work is of potential neuroscientific and methodological interest to researchers studying functional resting-state networks and brain development, as well as to deep learning scientists.

## Introduction

In utero fetal brain development follows a highly organized, dynamic, and precisely patterned process that lays a critical foundation for lifelong neurodevelopmental and neuropsychiatric brain health

(*Amgalan et al., 2021*; *De Asis-Cruz et al., 2022*). The architecture of the human connectome also changes rapidly with brain maturation and plays a critical role in critical periods of early brain development (*De Asis-Cruz et al., 2021b*; *De Asis-Cruz et al., 2021a*; *Gilmore et al., 2018*; *Cao et al., 2017b*).

Functional magnetic resonance imaging (fMRI) has become a powerful neuroimaging tool for investigating whole brain dynamics (*Fox and Raichle, 2007*). Seminal fMRI studies have shown that the mature human brain is comprised of several overlapping large-scale brain networks subserving different sensorimotor and higher cognitive functions (*Damoiseaux et al., 2006*; *Buckner et al., 2011*). These functional brain networks (FBNs) evolve over the life span, from infancy to old age (*Vij et al., 2018*; *Hu et al., 2022*; *Gao et al., 2009*; *Gao et al., 2015*). The recent successful application of resting-state fMRI (rsfMRI) to the human fetus and newborn has extended these critical observations and shed new light on the emergence of these FBNs at early life span. Sensorimotor networks have been shown to develop around the third trimester of pregnancy ~30 wk (*Thomason et al., 2015*), while higher cognitive networks emerge later in neurodevelopment (*Turk et al., 2019*). For example, the primitive form of the default mode network emerges around 35 wk in fetal (*Thomason et al., 2015*; *Thomason et al., 2014*) or newborn (*Doria et al., 2010*) rsfMRI and highly resembles the adult configuration by the end of the first year (*Gao et al., 2015*). However, to date, analysis of fetal and neonatal rsfMRI data has heavily relied on linear models, using heuristic knowledge learned from adult rsfMRI studies such as brain atlases (*Turk et al., 2019*), brain network patterns (*Xu et al., 2019*), or driven by data-driven linear computational models such as independent component analysis (ICA) (*Eyre et al., 2021*). In the regime of linear analytical models, complex rsfMRI data (e.g., no. of vertex >50,000) is considered as a linear combination of timeseries of several representations (e.g., no. of cortical parcel = 360 or no. of IC components = 50–300). While modeling adult rsfMRI activity as a linear system has been shown to be a simple and reasonable approach (*Logothetis et al., 2001*; *Schölvinck et al., 2010*; *Mukamel et al., 2005*) (see review *Boynton et al., 2012*), the utility of linear models in fetal– neonatal fMRI analysis can be limited due to the rapid neurodevelopment occurring during the fetal–neonatal stage (*Andescavage et al., 2017*; *Clouchoux et al., 2012*; *Li et al., 2021*).

It is now recognized that the relation between hemodynamic function linking neural activity to the measured fMRI signal is nonlinear (*Friston et al., 2000*; *Vazquez et al., 2006*; *Wager et al., 2005*). Consequently, recent studies with nonlinear modeling of fMRI data have yielded more informative brain representations compared to conventional linear methods, leading to significant improvements in disease classification (*Su et al., 2013*) and identifying individual-specific (i.e., as opposed to group-averaged) brain features based on functional connectivity (FC) (*Kim et al., 2021a*), among others. In addition to the nonlinearity inherent in fMRI measurement, studies have shown that regions of the brain mature at different rates (*Doria et al., 2010*; *Eyre et al., 2021*; *Cao et al., 2017a*). For example, fetal inter-hemispheric long-range connectivity has an inflection point between 26 and 29 wk when connectivity strength rapidly increased in a region-specific manner (i.e., occipital before temporal before frontal region), and then plateaus at around 32 wk (*Jakab et al., 2014*). Taken together, these findings suggest that investigation of fetal–neonatal FBNs using rsfMRI would likely require a computational approach that could potentially capture the nonlinearity inherent to rsfMRI data into the model. To the best of our knowledge, most investigations of emerging FC have utilized linear models, which may fail to capture important nonlinear information of the highly dynamic fetal–neonatal brain continuum that is critical for more accurate characterization of functional brain development in the early stages of life.

Recently, we developed a novel nonlinear analysis framework for adult rsfMRI using unsupervised deep generative model – variational autoencoder (VAE) (*Kim et al., 2021a*). Different from linear analytical models that decomposes complex, multivariate rs-fMRI data into independent, additive components, our VAE model learns a nonlinear feature set (or 'latent space') effectively using large-scale adult rsfMRI data through several nonlinear layers embedded in the VAE model. Our preliminary results in adults (*Kim et al., 2021a*) demonstrated that the fully trained VAE model could disentangle generative factors of rsfMRI data and encode the learned representations as latent variables. Noteworthy, generated VAE representations in the latent space were robust over varying signal quality of rsfMRI. Given the initial success, we posited that the VAE could directly translate nonlinear information learnt from adult rsfMRI to nonlinearity of unseen fetal–neonatal rsfMRI. It motivated us to apply the VAE model to characterize in vivo fetal–neonatal brain maturation and to critically investigate

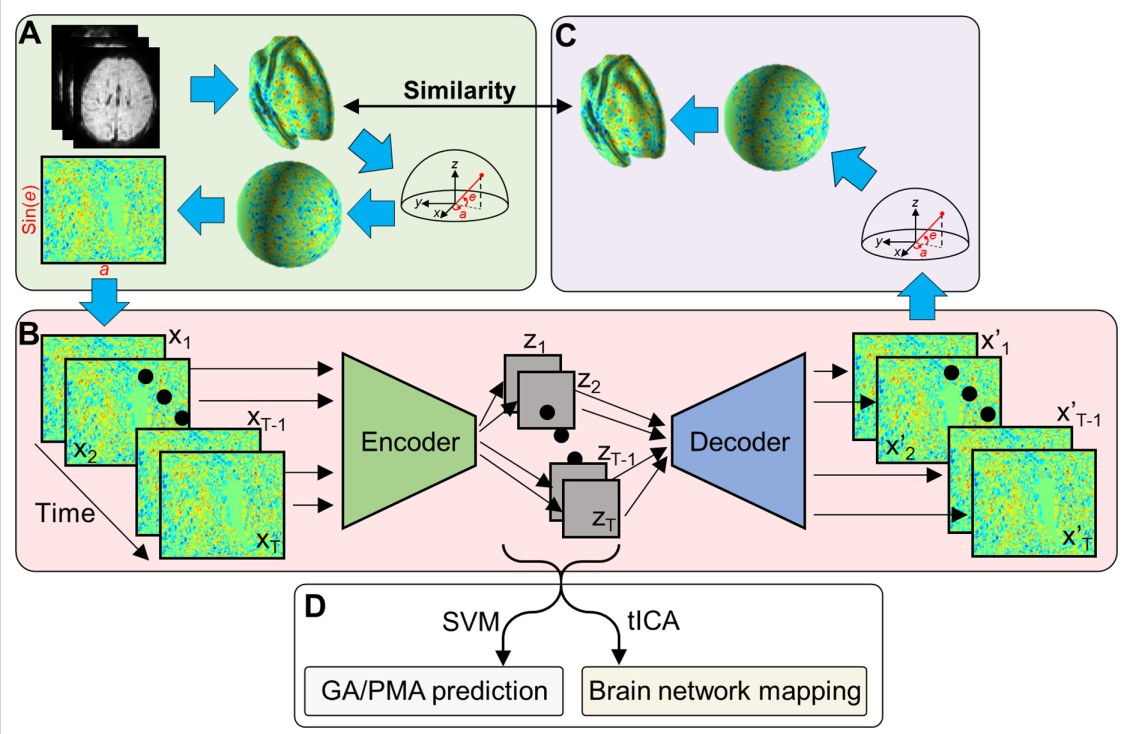

**Figure 1.** Variational autoencoder (VAE) and its application on fetal–neonatal functional magnetic resonance imaging (fMRI) data. (**A**) From volumetric fMRI patterns, 2D brain pattern is estimated via geometric reformatting. (**B**) Through the encoder and decoder of VAE, each brain pattern is compressed to latent variable z and reconstructed to x'. (**C**) Reconstructed 2D image is re-shaped into cortical space through the inverse reformatting step. (**D**) Latent representations estimated by VAE are used as features of different analysis.

the nonlinearity of fetal–neonatal rsfMRI data. We hypothesized that nonlinear representations of fetal–neonatal rsfMRI, extracted using the VAE model pretrained with the rsfMRI of healthy adults, would carry more accurate and informative neural signatures of the prenatal-neonatal FBNs, compared to linear representations defined at the network scale using ICA or at the regional scale using multi-modal cortical parcel. To validate our hypothesis, we conducted two experiments: first, compression of instantaneous fMRI patterns of subjects; and second, prediction of gestational/postmenstrual ages (PMAs) using their rsfMRI scans. The generalizability of our findings was carefully tested using two large datasets acquired by different institutions, together consisting of >500 fetuses, premature and healthy term-born neonates. Finally, using latent variables at the group level, we mapped fetal–neonatal brain resting-state networks at different age groups.

## Results

### VAE represented fetal–neonatal fMRI patterns better than linear models

We first determined whether our VAE model pretrained using adult rsfMRI data could extract meaningful representations of fetal–neonatal cortical activity. To accomplish this, we measured the reconstruction performance of rsfMRI patterns via the VAE model and compared it to linear representations derived using ICA and cortical parcels. For the latter, we employed linear latent spaces defined through group ICA of adult rsfMRI, the same dataset used for VAE training (see 'Materials and methods' for details). As the number of dimensions of latent space (i.e., number of features) is critical to reconstruction performance, we utilized linear latent spaces with different dimensions (i.e., IC50, IC100, IC200, and IC300; publicly available at https://db.humanconnectome.org/; *Van Essen et al., 2013* and melodic ICA with 256 components). Reconstruction performance was defined as the spatial correlation between original and reconstructed cortical patterns (*Figure 1A–C*).

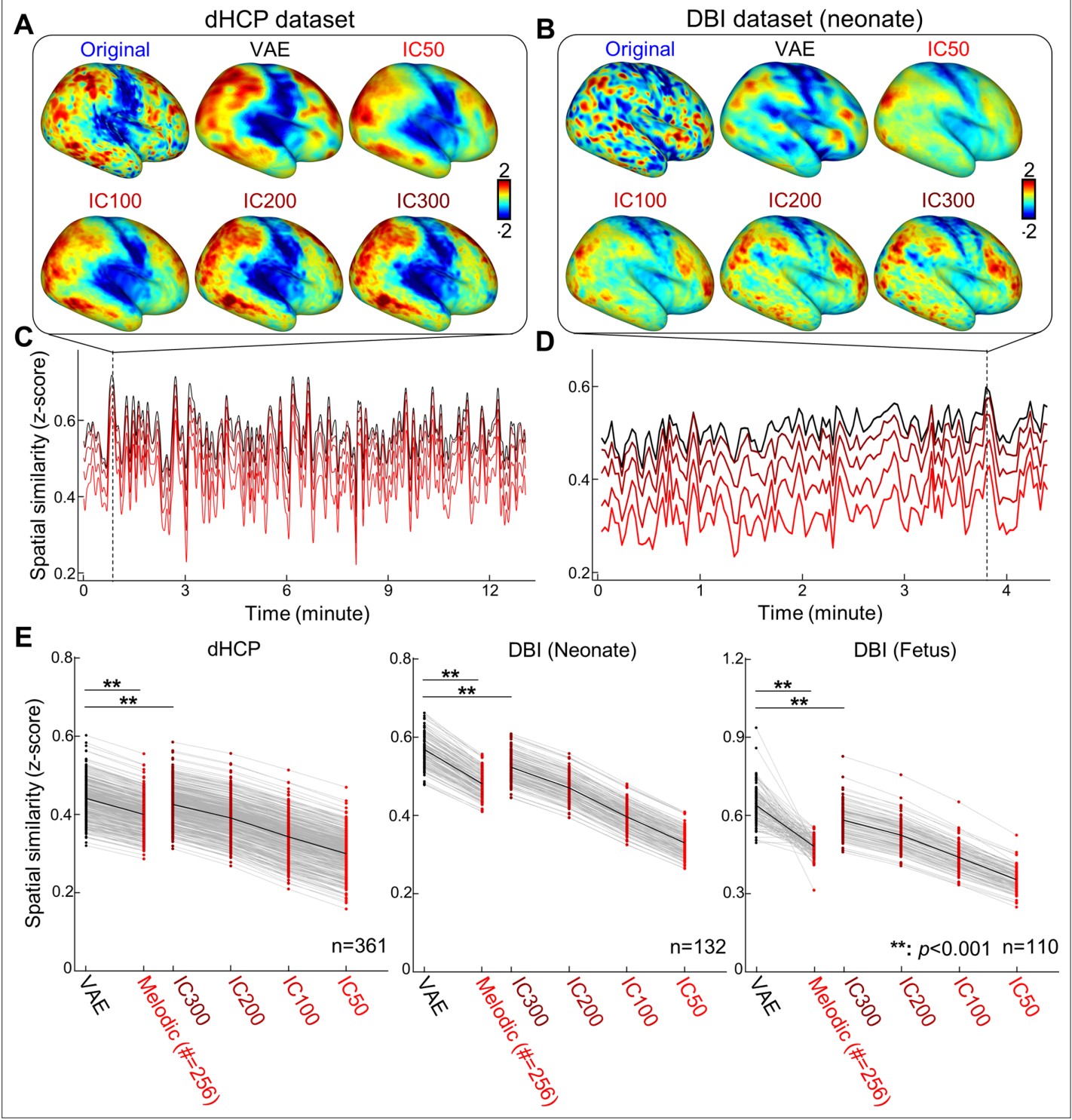

**Figure 2.** Variational autoencoder (VAE) represented fetal–neonatal functional magnetic resonance imaging (fMRI) patterns better than linear counterparts. Compared to linear spaces defined by group independent component analysis (IC50, IC100, IC200, and IC300), VAE shows the best reconstruction performance on both Developing Human Connectome Project (dHCP) (**A, C**) and Developing Brain Institute (DBI) datasets (**B, D**), at the individual level and at the group level (**E**). **p<0.001, Bonferroni-corrected.

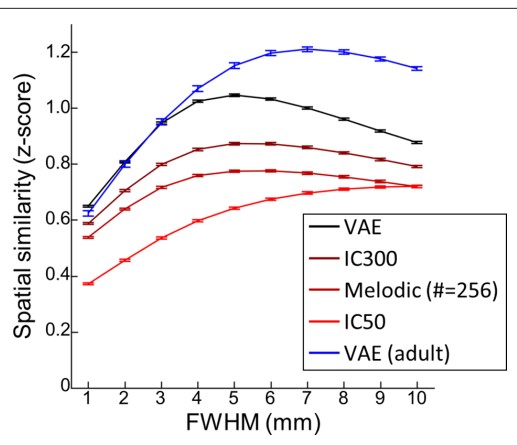

**Figure 3.** Smoothing effect of variational autoencoder (VAE) and linear counterparts. A function of the full-half-maximum-width (FWHM) (from 1 to 10 mm) used for spatial smoothing of Developing Brain Institute (DBI) neonate functional magnetic resonance imaging (fMRI) images. The error bar stands for the standard error of mean. n = 139.

Qualitatively, we observed that the VAE model successfully reconstructed the fetal–neonatal cortical patterns but at the cost of some smoothing effects (*Figure 2A and B*). While linear latent spaces with different dimensions 50, 100, 200, and 300 (IC50-300) also exhibited reasonable reconstruction performance, VAE showed more accurate reconstruction patterns compared to the original image, especially in the frontal (*Figure 2A*) and temporal regions (*Figure 2B*). To confirm our qualitative observation, we quantified the spatial similarity between original and reconstructed brain activity patterns, for each time point (*Figure 2C and D*). Although spatial similarity varied over time course, VAE always showed the best similarity compared to other linear counterparts. This observation remained consistent at the group level and across different datasets (*Figure 2E*). Interestingly, IC maps with higher (=300) dimensions than VAE (=256) showed inferior reconstruction performance than VAE model (*Figure 2E*). To eliminate possible bias due to different number of components in the comparison, we derived IC map with 256 components (name as melodicIC) from the human adult rsfMRI data that was identically used in the VAE training and found that melodicIC showed inferior reconstruction performance compared to the VAE or IC300. We further confirmed that the difference in reconstruction performance between VAE and IC300 was statistically significant (paired *t*-test, Bonferroni-corrected $p < 10^{-3}$), for all datasets.

To test whether improved representations in the VAE stemmed from a stronger smoothing effect of models, we evaluated the smoothing effects of the VAE and linear models as a function of varying full-half-maximum-width (FWHM) level (*Figure 3*). Briefly, we measured spatial similarity between reconstructed and manually smoothed activity patterns at different FWHM levels. The smoothing effect of the models was approximated by its peak spatial similarity (e.g., smoothing effect of the VAE = 5 mm FWHM). If better reconstruction performance of the VAE model merely originated from its stronger smoothing effect, the trends of spatial similarity of the VAE and linear models over increasing FWHM levels would intersect or cross over. However, this was not observed, suggesting that the improved VAE reconstruction performance of the VAE stemmed from its superior representations of neonatal rs-fcMRI.

As expected, reconstruction performance of adult rsfMRI (blue line) was better than neonatal rsfMRI data. Also, the maximum FWHM level was at 7 mm for adults compared to 5 mm for neonates. Given that brain size of newborns is about 60% of adults' brain size, this result suggested that smoothing effect of the VAE is largely consistent against age-related characteristics of rsfMRI. Similar smoothing effect was observed for IC300 and melodic IC with 256 components. IC50 showed the strongest smoothing effect as the peak was beyond our FWHM search boundary. Nonetheless, the VAE showed the best reconstruction performance compared to other linear counterparts across all smoothing levels.

## Reconstructed fMRI patterns contained not only age, but also individual neural signatures

We speculated that a critical factor of subject-wise variation in the reconstruction performance was the variability in participants' ages at the time of the MRI study. We found that PMA at scan was significantly and positively correlated with the reconstruction performance for both Developing Human Connectome Project (dHCP) (*Figure 4A*; r = 0.20, p<0.01) and Developing Brain Institute (DBI) (*Figure 4B*; r = 0.35, p<0.01) datasets. When only scans with good radiology scores (=1 or 1 + 2) were considered, the association between age at scan and reconstruction performance remained

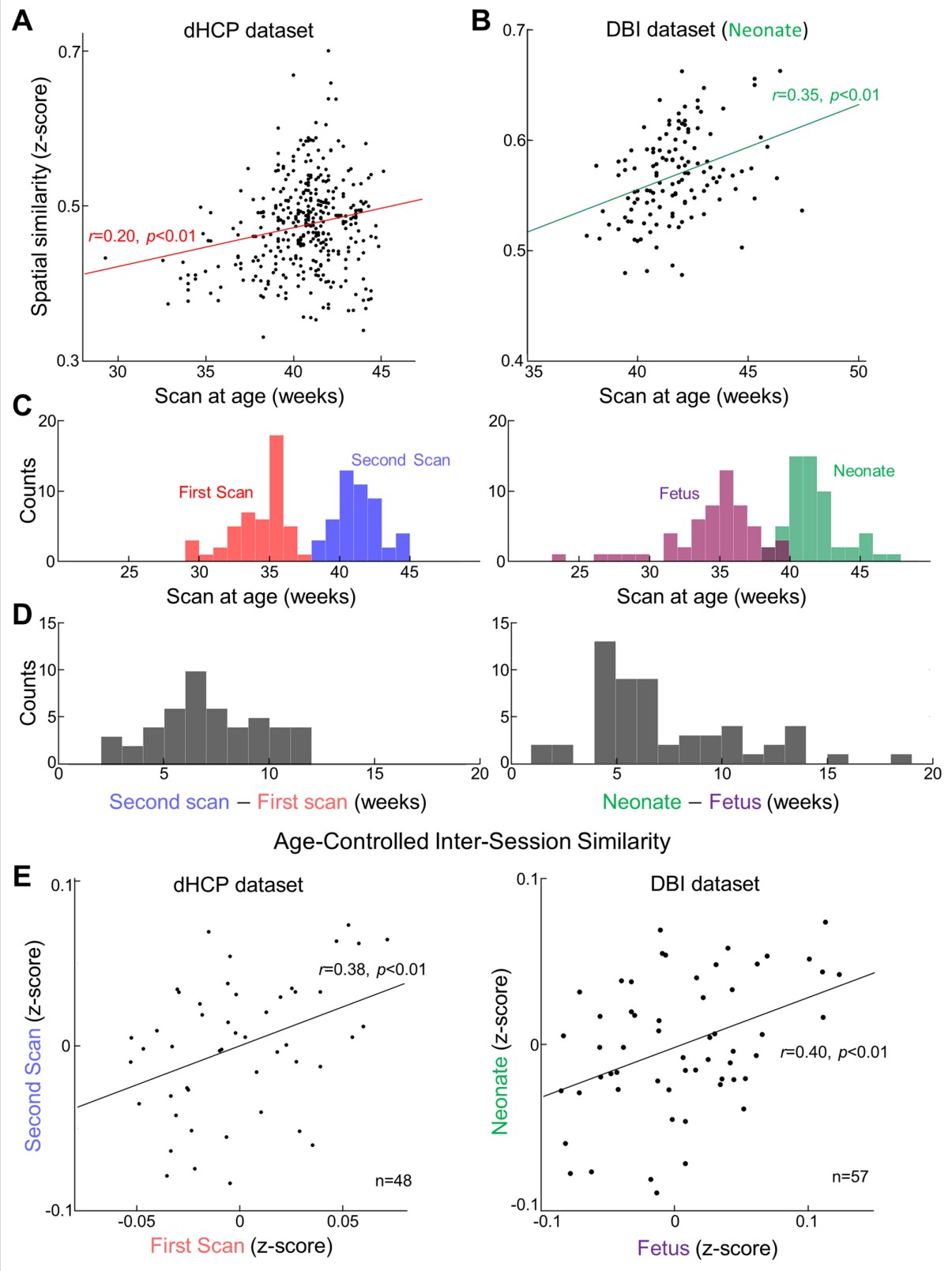

**Figure 4.** Reconstruction degree of functional magnetic resonance imaging (fMRI) patterns not only varies across different ages at scan but also is individual-specific. Reconstruction degrees across individuals are positively correlated with their ages at scan for both datasets, Developing Human Connectome Project (dHCP) (**A**) and Developing Brain Institute (DBI) (**B**). In (**B**), each red dot stands for the outliers having >3 median absolute deviations away from the median. (**C**) Distributions of ages at repeated scans (left; dHCP dataset, first and second scans; right; DBI dataset, fetal and

*Figure 4 continued on next page*

*Figure 4 continued*

neonatal scans). (**D**) Distributions of inter-scan age difference, for dHCP (left) and DBI dataset (right). (**E**) Scatterplot of reconstruction degree between the first scan and (or fetal scan for DBI) and the second scan (or neonatal scan for DBI), after controlling the effect from ages at scan.

The online version of this article includes the following figure supplement(s) for figure 4:

**Figure supplement 1.** Smoothing effect of young fetal brain during projecting to standard brain template.

**Figure supplement 2.** Age-dependency of reconstruction performance is not driven by scan length.

**Figure supplement 3.** Reconstruction variability among individuals is not related to head motion artifact.

significant (group with radiology score = 1, normal appearance for age; n = 172, $r$ = 0.19, p=0.01; score = 1 + 2, 2 = incidental findings with unlikely significance for clinical outcome or analysis; n = 292, $r$ = 0.21, p<$10^{-3}$). Conversely, for the fetal dataset, we found that the reconstruction performance was negatively correlated with GA (**Figure 4—figure supplement 1C**; $r$ = −0.69, p<$10^{-4}$). We posit that this was in part due to the smoothing effect induced when registering the small size of the fetal brain (e.g., 19 wk) to the standard 40 wk neonatal brain template, for example, the brain size of fetus at GA = 19 wk was only 52 cc compared to 330 cc in fetus with GA = 32 wk (**Figure 4—figure supplement 1A**). Therefore, projecting fMRI activity in smaller brain into the standard brain template introduced greater ballooning effect to brain activity; this was somewhat akin to applying heavy spatial smoothing procedure to the activity pattern (see **Figure 4—figure supplement 1B**). Similarly, we observed a strong negative association between brain size and reconstruction performance (**Figure 4—figure supplement 1D**). Once brain size was accounted for, the negative correlation between age and reconstruction performance was reduced (**Figure 4—figure supplement 1E**, $r$ = −0.24, p=0.01). However, despite the reconstruction performance being reduced, the negative correlation between fetal age and reconstruction performance remained significant. While we posit that brain representations of younger fetuses are less similar to human adults than older fetuses, our observation suggests that there are additional confounding factors that we failed to identify. In contrast, brain size in newborns was not significantly associated with reconstruction performance (**Figure 4—figure supplement 1G**, $r$ = 0.14, p=0.12). We believe this is because brain sizes were less variable in neonates (mean = 465.25 cc vs. STD = 58.4 cc) compared to fetuses (mean = 340.88 cc vs. STD = 86.90 cc) and because the standard brain template lay within the age range of the neonates. For these reasons, it is likely that the smoothing effect induced by the brain size is negligible in the neonates but not in the fetuses. Lastly, the positive age dependency of reconstruction performance in neonates remained significant even when considering brain size (**Figure 4—figure supplement 1F**, $r$ = 0.36, p<$10^{-4}$). Further studies investigating the influence of brain size and reconstruction performance in very preterm infants may help clarify the factors driving the discrepancy in findings between fetal- and neonatal groups.

Interestingly, we found that the age dependency of the reconstruction performance was significant, but the strength of correlation was moderate $r$ = 0.20 (p<0.01; **Figure 4A**) and $r$ = 0.35 (p<0.01; **Figure 4B**), in the dHCP and DBI datasets, respectively. In the dHCP dataset, reconstruction performance was negatively correlated with average frame-wise displacement level ($r$ = −0.34, p<$10^{-4}$) but not with brain size ($r$ = −0.07, p=0.17). There was no significant correlation between PMA at scan and head motion ($r$ = 0.01, p=0.79). When head motion level was considered in the model, the age dependency of the reconstruction performance increased ($r$ = 0.22, p<$10^{-4}$). The reconstruction performance remained consistent across different scan duration (**Figure 4—figure supplement 2**).

We sought to determine whether reconstruction performance in newborns was consistent across scans. To investigate this, we analyzed the reconstruction performance in subjects with two postnatal scans (dHCP dataset; n = 48 subjects; **Figure 4C**, left) or those with in and ex utero scans (DBI dataset; n = 57 subjects; **Figure 4C**, right). The time interval between scans for the dHCP dataset was 3–12 wk (**Figure 4D**, left) and 2–19 wk (**Figure 4D**, right) for the DBI datasets. After regressing out the age effect from the reconstruction performance, we found that subjects having better reconstruction performance on their first scan also tended to have better reconstruction performance on their second scan (**Figure 4E**, left; $r$ = 0.38, p<0.01; intraclass correlation coefficient [ICC] = 0.38). Interestingly, this trend was also observed in subjects with in utero and ex utero scans (**Figure 4E**, right; $r$ = 0.40, p<0.01; ICC = 0.39). To confirm that this finding was not driven by similar head motion profiles between scans, we analyzed whether head motion was related between the subjects' two scans. We found that there was no significant correlation in the inter-session frame-wise head displacement

(*Figure 4—figure supplement 3*, left; *r* = 0.15, p=0.32). The inter-session similarity of reconstruction performance also remained significant after controlling for head motion (*Figure 4—figure supplement 3*, middle; *r* = 0.30, p<0.05). We further found the inter-session similarity of reconstruction performance was not driven by the variation in brain sizes for both dHCP and DBI datasets (for DBI dataset, *Figure 4—figure supplement 1H*; *r* = 0.36, p<0.01, and for dHCP dataset, *Figure 4—figure supplement 3*, middle; *r* = 0.37, p<0.01). Altogether, our results suggest that the individual variation of the reconstruction performance of resting-state fetal–neonatal brain activity was tightly connected not only to chronological brain maturity but also to the intrinsic neural traits of the subjects even during the in utero period.

## Prediction of gestational age and postmenstrual age using VAE-derived representations

Next, we investigated whether latent variables extracted by VAE contained neural signatures of fetal–neonatal functional brain maturation using the dHCP dataset. Given different types of latent variables that were defined by VAE, cortical parcels, or IC50-300, we calculated FC patterns, which

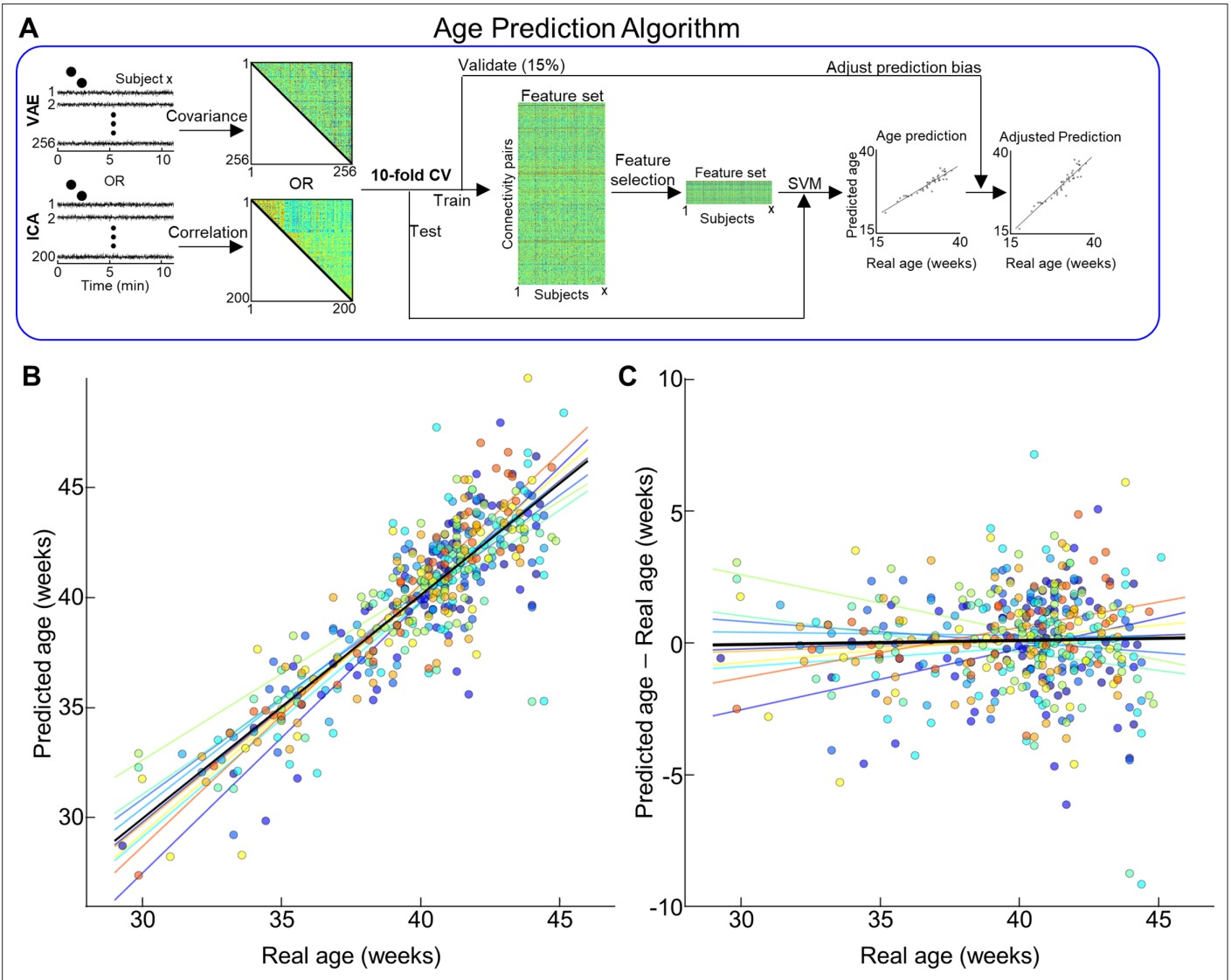

**Figure 5.** Age prediction based on different latent representations of neonates in the Developing Human Connectome Project (dHCP) dataset. (**A**) Illustration of age prediction algorithm. (**B**) Scatterplot between actual age and predicted age. (**C**) Distribution of prediction error across age at scan. (**B, C**) Different colors stand for the prediction age from different folds. Lines with different colors stand for the optimal fit. Black line is the optimal fit for whole samples.

**Table 1.** Comparison of age prediction performance in Developing Human Connectome Project (dHCP) dataset using different latent representations.

RMSE: root mean squared error; MAE: mean absolute error; $R^2$: explained variance; VAE: variational autoencoder; ICA: independent component analysis. Red highlight indicates the best performance among different latent representation methods.

| Representation method | RMSE (mean ± SD) | MAE | $R^2$ | Correlation |
|---|---|---|---|---|
| VAE (N = 256) | 1.98 ± 0.06 | 1.53 ± 0.05 | 0.72 ± 0.01 | 0.84 ± 0.02 |
| Cortical parcel (N = 360) | 2.35 ± 0.08*** | 1.85 ± 0.07*** | 0.65 ± 0.01*** | 0.80 ± 0.02*** |
| IC50 | 2.88 ± 0.15*** | 2.28 ± 0.12*** | 0.55 ± 0.02*** | 0.73 ± 0.03*** |
| IC100 | 2.67 ± 0.11*** | 2.12 ± 0.09*** | 0.59 ± 0.02*** | 0.76 ± 0.03*** |
| IC200 | 2.31 ± 0.09*** | 1.83 ± 0.07*** | 0.65 ± 0.01*** | 0.80 ± 0.02*** |
| IC300 | 2.43 ± 0.09*** | 1.94 ± 0.07*** | 0.63 ± 0.01*** | 0.79 ± 0.02*** |
| Melodic ICA (N = 256) | 2.23 ± 0.08*** | 1.76 ± 0.07*** | 0.67 ± 0.01*** | 0.81 ± 0.02*** |

***: Bonferroni-corrected $p<10^{-4}$; compared to VAE.

was defined by the covariance (or correlation for cortical parcels and IC50-300) for every pair of latent variables, per subject (*Figure 5A*). Through fusing tenfold cross-validation scheme and linear regression support vector machine (RSVM) in dHCP dataset, we found that latent variables derived by VAE yielded highly reliable prediction accuracy (*Figure 5B and C*). As suggested in brain age prediction studies (*Gong et al., 2021*), we also adjusted for prediction bias in the model (*Figure 5C*). Interestingly, the age prediction model with VAE-derived representations reached the best age prediction accuracy; mean ± standard deviation over tenfolds, root mean squared error (RMSE) = 1.98 ± 0.06, mean absolute error (MAE) = 1.53 ± 0.05, $r^2$ = 0.72 ± 0.01, with a big margin compared to models based on linear latent features (*Table 1*). The percentage (=14.14 ± 0.32%) of latent space representing age information was lower than most of the linear counterparts (for cortical space, 76.53 ± 1.29%; for IC50, 44.84 ± 0.46%; for IC100, 32.55 ± 0.43%; for IC200, 16.07 ± 0.34%; for melodicIC, 29.97 ± 0.47%), except IC300 (=10.83 ± 0.25%). It is noteworthy that different from the monotonic improvement of the reconstruction performance over increasing IC #, the age prediction accuracy reached peak at melodicIC while accuracy deteriorated in the prediction model with features from 300 ICs (melodicIC: 1.76 ± 0.07 vs. 1.94 ± 0.07 for IC300). The age prediction accuracy of the cortical

**Table 2.** Comparison of age prediction performance in Developing Brain Institute (DBI) dataset using different latent representations.

RMSE: root mean squared error; MAE: mean absolute error; $R^2$: explained variance; VAE: variational autoencoder; ICA: independent component analysis. Red highlight indicates the best performance among different latent representation methods.

| Representation method | RMSE (mean ± SD) | MAE | $R^2$ | Correlation |
|---|---|---|---|---|
| VAE (N = 256) | 3.83 ± 0.20 | 3.02 ± 0.17 | 0.66 ± 0.01 | 0.79 ± 0.03 |
| Cortical parcel (N = 360) | 4.55 ± 0.27*** | 3.64 ± 0.22*** | 0.58 ± 0.01*** | 0.73 ± 0.02*** |
| IC50 | 5.80 ± 0.52*** | 4.65 ± 0.43*** | 0.48 ± 0.02*** | 0.65 ± 0.04*** |
| IC100 | 5.74 ± 0.33*** | 3.78 ± 0.28*** | 0.57 ± 0.02*** | 0.73 ± 0.04*** |
| IC200 | 5.05 ± 0.36*** | 3.97 ± 0.29*** | 0.54 ± 0.02*** | 0.70 ± 0.03*** |
| IC300 | 4.82 ± 0.33*** | 3.79 ± 0.27*** | 0.57 ± 0.02*** | 0.72 ± 0.03*** |
| Melodic ICA (N = 256) | 4.24 ± 0.32*** | 3.33 ± 0.27*** | 0.62 ± 0.01*** | 0.76 ± 0.03*** |

***: Bonferroni-corrected $p<10^{-4}$; compared to VAE.

parcels (MAE: 1.85 ± 0.07) model was between the IC300 and melodicIC models. Collectively, these results suggest that nonlinear features extracted by VAE improved age prediction, outperforming linear features derived using ICA or cortical parcels.

We also tested our age prediction scheme under different latent representations methods in the DBI dataset. Unlike the dHCP dataset, fetuses and neonates in the DBI dataset had fMRI scans with different spatiotemporal resolutions. To minimize possible effects of this difference, we modified the feature selection step in the prediction scheme, deriving the global network strength by summing significant and positive FC edges per subject, yielding a single feature per subject. As the feature space became 1, we substituted linear regression for RSVM. In line with findings in the dHCP dataset, the results clearly showed that latent variables derived by VAE reflected their neurodevelopmental variability across fetuses and neonates more effectively than linear latent representations (*Table 2*). When the age prediction model considered a single age group (neonates or fetuses), the age prediction accuracy of VAE was superior to linear models only in neonates; cortical parcel showed better performance in fetuses (*Supplementary file 1*). The percentage of VAE-derived latent space representing age information (=11.39 ± 0.35%) was lower than IC100 (=10.86 ± 0.27%), IC200 (=7.31 ± 0.25%), and IC300 (=5.24 ± 0.23%); comparable to cortical parcels (=11.12 ± 0.24%); higher than IC50 (=21.68 ± 0.43%) and melodicIC (=12.6 ± 0.24%).

## VAE-derived representations showed better cross-center generalizability of age prediction than linear representations

Clinical utility of an age prediction model relies on its generalizability across datasets acquired under different data acquisition parameters, for example, spatial and temporal resolution, SNR level, etc. For this purpose, we tested the inter-center generalizability of our age prediction model over dHCP

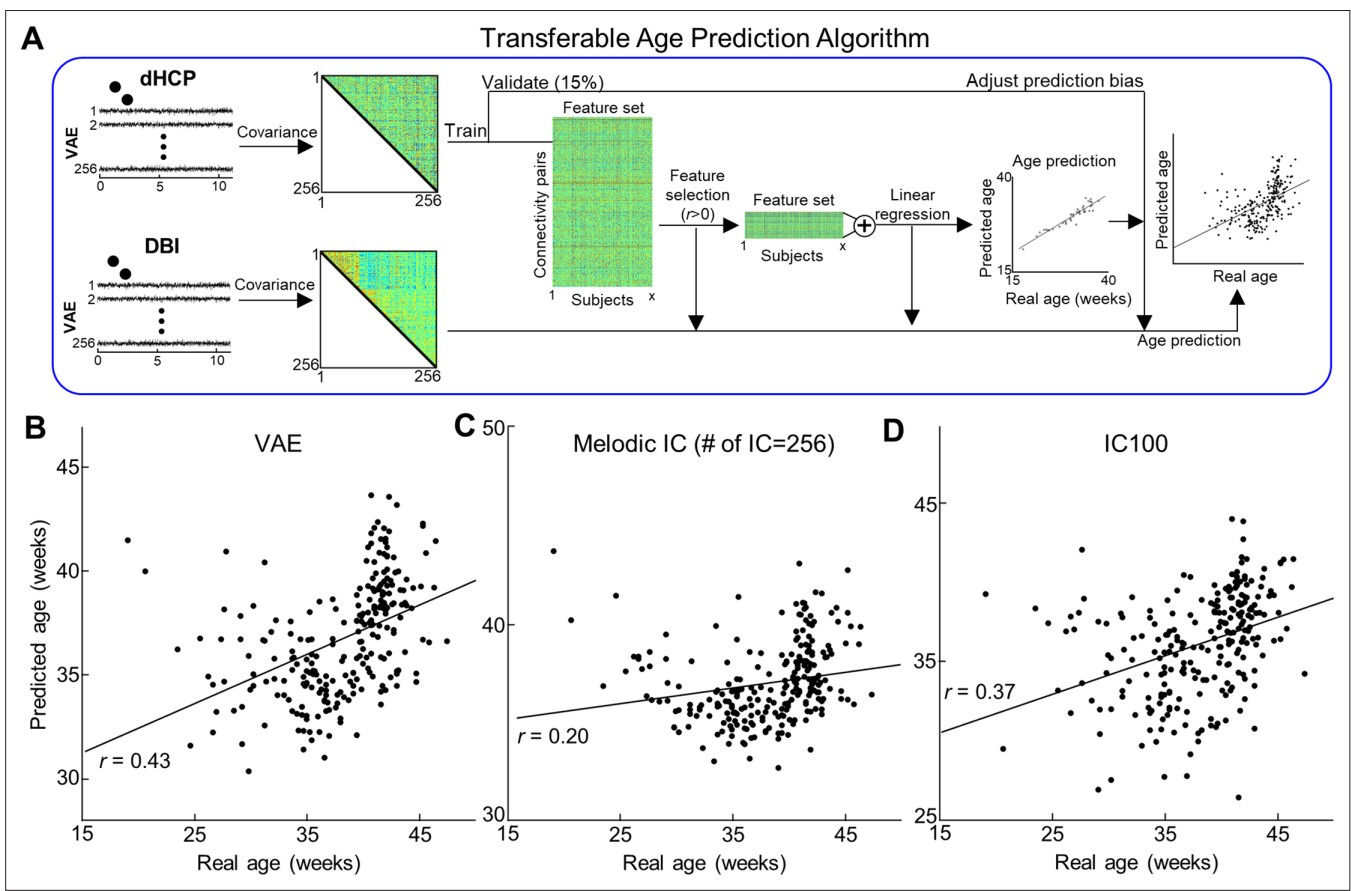

**Figure 6.** Cross-center age prediction algorithm using different latent representations. (**A**) Illustration of transferable age prediction algorithm. (**B–D**) Scatterplot between actual age and predicted age using different latent representations. Each red dot stands for the outliers having >3 median absolute deviations away from the median reconstruction performance.

**Table 3.** Cross-center generalizability of age prediction performance under different latent representations.

RMSE: root mean squared error; MAE: mean absolute error; VAE: variational autoencoder; ICA: independent component analysis. Red highlight indicates the best performance among different latent representation methods.

| Representation method | RMSE | MAE | vs. VAE (MAE) |
|---|---|---|---|
| VAE (N = 256) | 4.95 ± 0.66 | 3.90 ± 0.65 | |
| Cortical parcel (N = 360) | 12.82 ± 15.38 | 12.06 ± 15.35 | $p<10^{-6}$ |
| IC50 | 5.67 ± 0.54 | 4.46 ± 0.54 | $p<10^{-6}$ |
| IC100 | 5.58 ± 0.92 | 4.46 ± 0.85 | $p<10^{-6}$ |
| IC200 | 5.72 ± 0.89 | 4.75 ± 0.83 | $p<10^{-6}$ |
| IC300 | 5.68 ± 0.98 | 4.71 ± 0.93 | $p<10^{-6}$ |
| Melodic ICA (N = 256) | 5.78 ± 1.62 | 4.72 ± 1.60 | $p<10^{-6}$ |

and DBI datasets using different latent spaces. As dHCP and DBI datasets had different MR sequence parameters, we re-employed the age prediction model used for the DBI dataset that summed significant and positive FC edges. Here, we used dHCP dataset as training and validation data, and tested the performance of trained models on the DBI dataset, as illustrated in *Figure 6A*. VAE showed the best cross-center prediction performance (*Figure 6B*) compared to linear models (*Figure 6C and D*). Quantitatively, we confirmed our observation that VAE showed the least error (RMSE = 4.95 ± 0.66, MAE = 3.90 ± 0.65; n = 100 random trials with bootstrapping permutation method) compared to other linear latent spaces (*Table 3*); cortical parcel and IC maps showed comparable prediction performance. Altogether, these results strongly support our hypothesis that nonlinear latent representations featured by the VAE convey neural signatures reflecting brain maturity across a wide range of gestational and PMAs, in a more effective way than linear latent representations.

## Mapping resting-state brain networks of neonatal babies using VAE

Here, we used nonlinear latent variables to map functional networks by utilizing both the VAE encoder and decoder. Briefly, the VAE encoder compressed each time point of rsfMRI cortical activity into each latent representation and the VAE decoder visualized latent representations in the cortical surface that we were interested in. As illustrated in *Figure 7A*, per subject, we estimated timeseries of latent variables by feeding each brain pattern to the VAE encoder. Estimated latent representations were concatenated across subjects. Then, we redefined 30 latent bases by applying temporal ICA to the concatenated latent variables. Finally, by feeding each independent latent basis into the VAE decoder, we visualized the cortical map of each independent latent basis (see details in 'Materials and methods'). Following this, we investigated FBNs for four groups (dHCP vs. DBI; fetus or preterm vs. full-term) separately. The age ranges of full-term in the dHCP and DBI datasets were 41.05 ± 1.76 and 41.71 ± 1.78, respectively. The ages of preterm in the dHCP dataset and fetus in the DBI dataset were 34.36 ± 1.85 and 33.65 ± 4.01, respectively. For a more optimal comparison between groups, we reordered the IC maps based on their pattern similarity by estimating Pearson correlation coefficients between every pair of independent latent bases, each from different groups (center matrix in *Figure 7B*).

Each independent latent basis estimated from the neonate group exhibited a unique FBN pattern consisting of both activation and deactivation patterns, as shown in *Figure 7B*. All FBN patterns for dHCP and DBI neonates are shown in *Figure 7—figure supplements 1 and 2*, respectively. For example, IC8 in both the dHCP and DBI datasets showed activation/deactivation patterns between left and right early visual area. This contrast (i.e., activation vs. deactivation) was also observed at the networks level; auditory vs. somatosensory (IC6 in dHCP; IC3 in DBI) and visual vs. auditory (IC3 in dHCP; IC4 in DBI). We also identified the neonatal auditory network having bilateral activation in IC3 for both the dHCP and DBI dataset. Lastly, we observed precursors of the default mode network (dHCP; IC29, DBI; IC21; *Figure 7B*, right bottom), which is coincident with findings in young adult

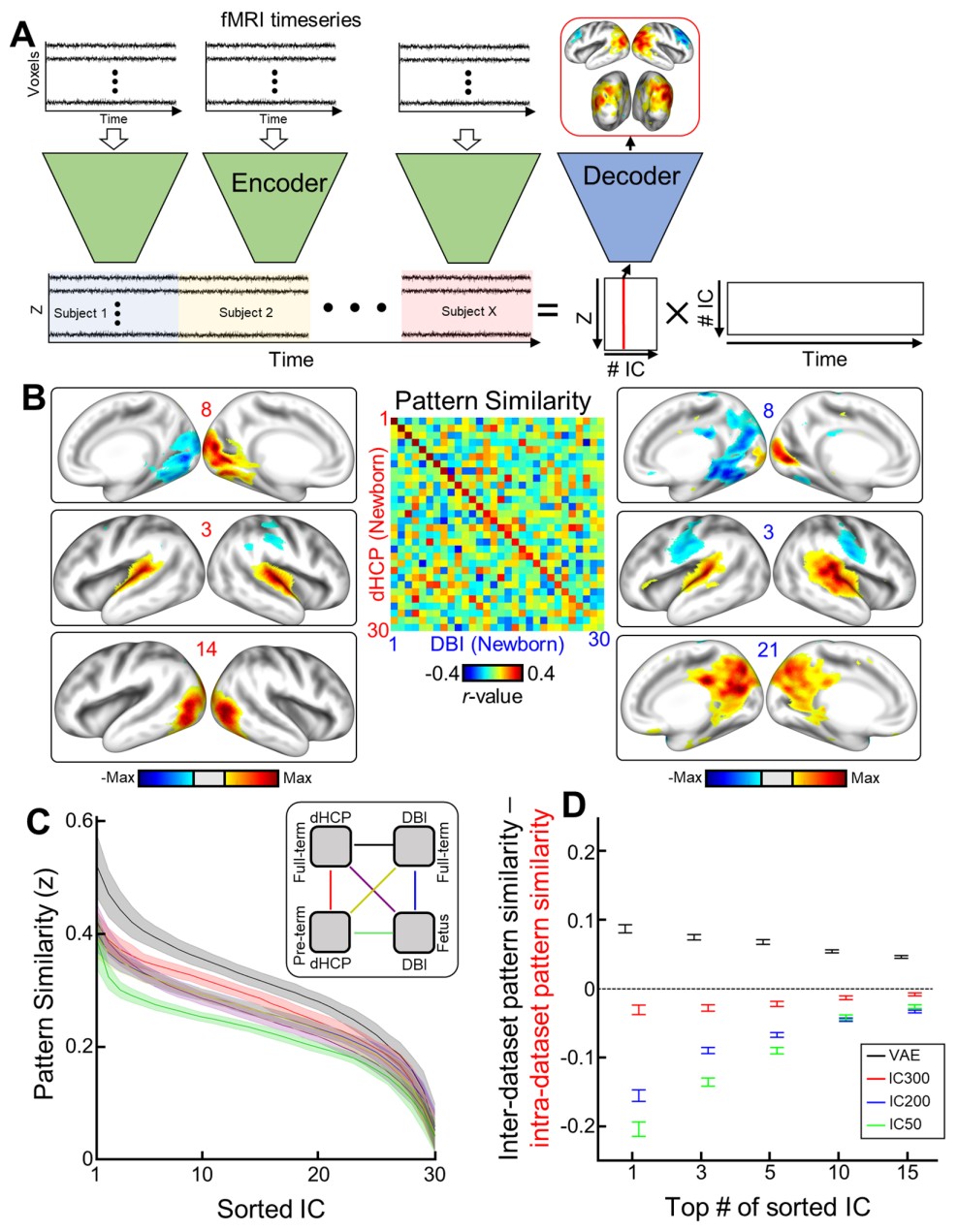

**Figure 7.** Mapping resting-state functional brain networks of fetuses and neonates using variational autoencoder (VAE). (**A**) Illustration describing how to map functional brain networks using the VAE. (**B**) Example of neonatal cortical networks estimated from the Developing Human Connectome Project (dHCP) dataset (left) or the Developing Brain Institute (DBI) dataset (right). The order of estimated independent latent variables is sorted by their absolute pattern similarity across different datasets (center). Each map is thresholded at the level of <15% of maximal absolute value. (**C**) Pattern similarities across different datasets and/or different age groups (coded as lines with different colors; right panel) are plotted. Shades and line stand for the standard deviation and mean similarity across 100 IC results with different initializations. (**D**) The difference between inter-dataset pattern similarity (neonate in DBI vs. dHCP) and intra-dataset pattern similarity (neonate vs. preemie in dHCP). The similarity was measured by averaging top 1–15 of sorted IC. Error bar stands for the standard deviation.

The online version of this article includes the following figure supplement(s) for figure 7:

**Figure supplement 1.** Full list of neonatal cortical networks estimated from dHCP dataset.

**Figure supplement 2.** Full list of neonatal cortical networks estimated from DBI dataset.

**Figure supplement 3.** Full list of cortical networks of preterm babies estimated from dHCP dataset.

*Figure 7 continued on next page*

**Figure supplement 4.** Full list of fetal cortical networks estimated from DBI dataset.

**Figure supplement 5.** Cortical pattern of IC3 in the neonatal dHCP dataset.

rsfMRI studies (*Eyre et al., 2021*; *Van Essen et al., 2013*; *Gong et al., 2021*). At the same time, there were brain network patterns that were observed only in the dHCP dataset. For example, we found inter-hemispheric opposition of the auditory network (IC22, *Figure 7—figure supplement 1*) and IC14 showed bilateral activation covering the higher visual network (*Figure 7B*, left bottom). Global synchrony patterns, which are tightly related to global signal in fMRI data, were observed in the dHCP dataset (IC30, *Figure 7—figure supplement 1*), but not in the DBI dataset (*Figure 7—figure supplement 2*). In the DBI dataset, we found simultaneous activation of somatosensory, auditory, and visual networks with deactivation of temporoparietal junction that integrates sensory inputs (IC6, *Figure 7—figure supplement 2*). However, some network patterns (e.g., IC5, 21, in the dHCP or IC16, 29, in the DBI) were less straightforward to interpret, given brain network patterns defined in human adults (*Beckmann et al., 2009*). FBNs of fetuses and preterm babies are shown in *Figure 7—figure supplements 3 and 4*, respectively. Lastly, we found that our observations were consistent regardless of the threshold level used (see an example FBN – IC3 from the dHCP dataset – at different thresholds, *Figure 7—figure supplement 5*).

To show that FBNs observed across different datasets and different age groups were not driven by noise inherent in data or artifact introduced during preprocessing steps, we examined the reproducibility of our findings in FBNs across different age groups and different datasets. We compared pattern similarity across different datasets and different age groups (*Figure 7C*). The best pattern similarity was achieved when their ages were matched (black line; full-term in dHCP vs. full-term in DBI), rather than between the age-unmatched groups from the same dataset (red line: full-term vs. preterm in dHCP; blue line: full-term vs. fetus in DBI). The least pattern similarity was observed between preterm neonates in the dHCP and fetuses in the DBI dataset. Similar results were reproduced against different number of ICs (#=1, 3, 5, 10, and 15) for calculating the pattern similarity. The opposite was observed with linear models. Specifically, the age-matched inter-dataset pattern similarity (black line in *Figure 7C*) was higher than intra-dataset similarity (red line in *Figure 7C*; full-term vs. preterm in dHCP) with VAE-derived latent variables. Linear models, on the contrary, showed higher intra-dataset similarity (*Figure 7D*). This observation remained consistent using varying number of top sorted ICs. The finding that the VAE revealed similar network patterns in two independent, age-matched neonatal datasets suggests that the VAE may be a better tool than linear models for capturing neurophysiologically relevant brain activity. In contrast, it is likely that ICA and cortical parcel techniques depended on features such as equivalent acquisition parameters, preprocessing, etc., that were similar between the pre- and full-term dHCP scans.

## Age-related group differences in VAE-defined networks

Lastly, to begin to understand how functional networks evolve as the brain matures during the newborn period, we investigated differences in nonlinear latent variables between young and old neonates. Briefly, we investigated whether the variance of each of the functional networks derived from the dHCP dataset correlated with advancing PMA at scan. Per subject, variance was estimated from the timeseries of the independent components (i.e., 30 functional networks). Of the 30 networks, the variance of 12 networks (IC2, 3, 6, 7, 10, 11, 16, 20, 21, 24, 25, 26, and 27) was significantly correlated with PMA at scan (FDR-corrected $p<0.05$; $r$ = 0.22, 0.29, 0.15,–0.16, –0.31, 0.16, 0.14, 0.17, 0.27, 0.16, 0.19, and 0.17, respectively). Among 12 functional networks, we further explored the age-dependent changes in IC6, the component most strongly correlated with PMA (*Figure 8C*; $r$ = 0.29, FDR-corrected $p<10^{-6}$). IC6 highlighted the joint of two networks, auditory and somatosensory; activation in somatosensory with deactivation in auditory (*Figure 8A*). We divided the sample into young (39 wk; n = 41) and old group (43 wk; n = 56) (*Figure 8B and C*). To set a relative baseline, we kept the remaining subjects as a median group (n = 287). As suggested by the correlational analysis, old group showed higher variance than younger group at IC6 (*Figure 8D*; uncorrected $p<10^{-7}$; see *Figure 8—figure supplement 1* for other networks) suggesting evolution of neonatal FBNs over aging. Estimated group-wise latent features were projected onto the cortical surface (*Figure 8E*). As expected, the functional network pattern from median group was very similar to that estimated from

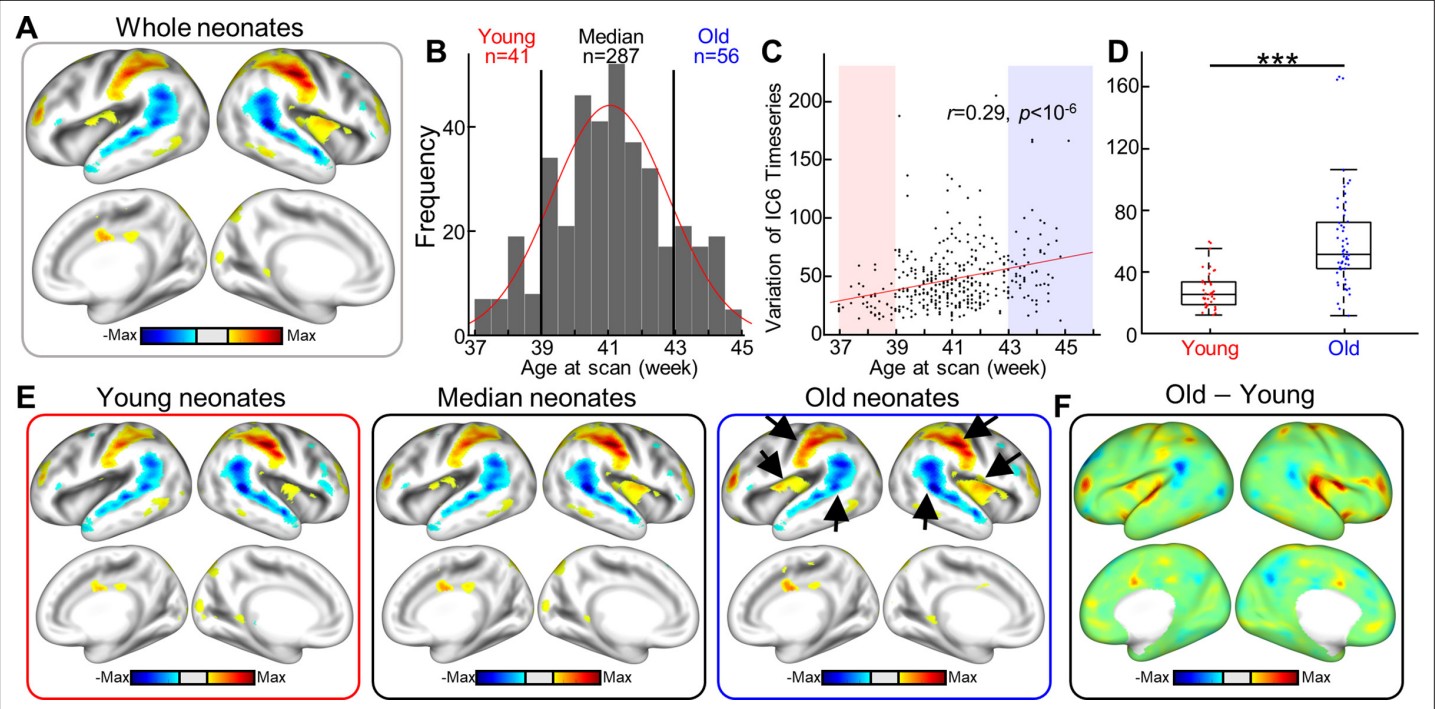

**Figure 8.** Between-group differences in resting-state functional brain networks using variational autoencoder (VAE). (**A**) The cortical network represents the sixth independent component (IC6) derived from all neonates of the Developing Human Connectome Project (dHCP). (**B**) The distribution of age at scan for the neonate cohort (n = 384). Based on the age distribution, the cohort was divided into three groups; young, median, and old groups. (**C**) Scatterplot between age at scan (x-axis) and IC6 timeseries variance. Each dot represents a subject; n = 384. (**D**) Variance boxplot in young (postmenstrual age [PMA] at scan 39 wk) and old groups (PMA at scan 43 wk) of the dHCP dataset. ***$p < 10^{-8}$. (**E**) Cortical networks in different age groups using dual-regression method. Inset arrows point to brain regions having different activation levels in older group compared to younger group. (**F**) Difference between two age groups; subtraction between groups is done at the latent space, followed by projection to the cortical surface.

The online version of this article includes the following figure supplement(s) for figure 8:

**Figure supplement 1.** Differences in resting-state functional brain networks (FBN) between young and old neonates.

**Figure supplement 2.** Group-wise difference in resting-state functional brain networks using VAE.

the entire cohort. To examine age-dependent change in FBNs, we subsequently compared old vs. young groups. In both groups, overall network pattern resembled the full sample's but there were notable regional differences in spatial involvement. For example, compared to the young group, the old group showed (1) mildly stronger activation in the somatosensory region, (2) stronger activation in the posterior insular region extending to the primary auditory region, and (3) weaker activation in the superior temporal auditory region (highlighted by black arrows in *Figure 8E*). To quantify these differences, we subtracted the functional networks of the two age groups (i.e., old–young) in the latent space and then projected differences onto the cortical space (*Figure 8F*). Functional networks of older groups showed stronger engagement of primary auditory area, with some level of lateralization toward the right hemisphere; no change in somatosensory region was observed. When subtraction was conducted at the cortical surface (i.e., comparison between FBNs of two groups), old neonates showed stronger activation in the somatosensory region along with engagement of the posterior insular region and primary auditory area (*Figure 8—figure supplement 2*). To summarize, our results demonstrate that the brain network engages additional brain region with advancing age (or maturity), suggesting the recharacterization of brain systems in the first weeks of the postnatal period.

## Discussion

Here, for the first time, we used VAE, an unsupervised deep learning model, to extract nonlinear representations in human fetal and newborn rsfMRI data. Compared to linear latent space defined by cortical parcels or ICA, nonlinear latent space defined by the pretrained VAE was superior in

representing rsfMRI patterns acquired using different MRI parameters (*Figure 2*). Noteworthy, we found that individual variability in reconstruction performance of brain activity patterns was consistent across repeated scans or between fetal and neonatal scans (*Figure 4*). We hypothesize that this may provide evidence for individual uniqueness embedded in the rs-fcMRI activity pattern as early as the neonatal period, but this requires further study. We further demonstrated that nonlinear representations extracted by using the VAE provided useful neural signatures of their brain maturity, yielding better age prediction in preterm and term neonates (or fetuses and neonates for the DBI dataset) compared to linear representations (*Figure 5*, *Tables 1 and 2*). By training and testing the prediction model with latent representations across different datasets, dHCP and DBI, respectively, we found that nonlinear representations showed the best inter-center generalizability for age prediction accuracy (*Figure 6* and *Table 3*). Finally, we identified 30 FBNs in neonates by applying temporal ICA to group-level nonlinear representations. Among the 30 brain networks, we identified sensory networks (visual, auditory, somatosensory), a precursor of default mode network (*Figure 7*, *Figure 7—figure supplements 1 and 2*), and complex patterns consisting of mixed brain networks (e.g., auditory- and visual networks together in IC3 and IC25; *Figure 7—figure supplement 1*). Finally, we successfully mapped the change of neonatal functional network for different age group (*Figure 8*). Collectively, our proposed VAE model and analytical scheme may have a potential to serve as an important complement to existing linear computational models for disentangling complex fMRI brain patterns in fetuses or neonates for the investigation of longitudinal neurodevelopment in healthy and high-risk fetal–neonatal populations.

In this study, VAE showed better reconstruction performance of fMRI patterns compared to ICA for each subject and across different age groups (*Figure 2E*). Interestingly, VAE outperformed ICA age prediction even when the dimensions of ICA (=300) latent variables exceeded that of the VAE (=256). We believe this stems from the ability of the VAE model to capture nonlinear features. In the computer vision field, deep learning studies *Russakovsky et al., 2015*; *Krizhevsky et al., 2012* have suggested that the nonlinear nature of deep convolutional models is a key for the model to extract meaningful features against morphological bias/changes of objects, for example, shift, rotation, or magnification in the image, yielding superior image classification performance. Like morphological distortions in natural images, fMRI images, despite efforts to spatially and temporally denoise the data, may still carry non-neural signals. Supporting this notion, we found that the percentage of VAE-derived latent variables sensitive to age prediction in dHCP dataset was comparable to the one observed in the DBI dataset (14.14 ± 0.32 vs. 11.39 ± 0.35%, for dHCP and DBI), whereas larger inter-dataset discrepancies were observed with linear models (most difference was in cortical parcel; 76.53 ± 1.29 vs. 11.39 ± 0.35% and least difference was in the IC300; 10.83 ± 0.25 vs. 5.24 ± 0.23%, for dHCP and DBI). In fact, when relatively narrower age variation was considered in the age prediction task (only fetus or neonate) in the DBI dataset, the VAE showed the best age prediction performance only in neonate; cortical parcel showed the best age prediction accuracy of fetal group. Nevertheless, the cortical parcel completely failed to predict the age across different datasets (MAE = 12.06 ± 15.35 wk), whereas VAE showed the best generalizability of age prediction across different datasets, MAE = 3.90 ± 0.65 wk (*Figure 6* and *Table 3*). Collectively, we posit that a cascade of nonlinear operations, efficiently implemented through several layers in our designed VAE architecture, disentangles complex brain activity patterns from residual nonlinear noise or artifact, differentiating the VAE model from linear models such as cortical parcel or ICA.

Recent neuroimaging studies have shown that FC patterns in individuals carry distinct signatures that are distinguishable from others (*Horien et al., 2019*; *Finn et al., 2015*). This unique pattern, according to a recent study, remained stable over time for about 3 years (*Horien et al., 2019*). However, the timing of when these unique individual features emerged is largely unknown. One neonatal study (*Ciarrusta et al., 2021*) that explored individual uniqueness of brain FC pattern by identifying individuals based on their FC profiles showed nearly no self-similarity of FC patterns between repeated scans (=11%) (*Ciarrusta et al., 2021*), suggesting that the functional uniqueness of the human brain likely emerges after the newborn period. In contrast, another study suggested that individual uniqueness was already apparent during the newborn period reaching an accuracy of 100% (n = 40) (*Wang et al., 2021*). However, in this study, data was acquired in a single session, split into two, then used to perform individual identification. This contrasts with other studies that used inter-session similarity to determine uniqueness of FC profiles. Thus, the perfect accuracy was likely largely driven by

measurement artifact as suggested in *King et al., 2023*. Compared to newborn data *Ciarrusta et al., 2021*, adults (~92%) *Finn et al., 2015*, children/youth (6–21 years old, ~43%) *Horien et al., 2019*, and infants (~1 year old, ~70%) *Hu et al., 2022* showed higher inter-scan similarity. In keeping with these data, we observed significant self-similarity – defined as the degree of similarity of reconstructed brain data – between repeated ex utero scans (*r* = 0.38, *Figure 4E*, left) and longitudinal in utero and ex utero scans (*r* = 0.39, *Figure 4E*, right). As the VAE was trained to represent resting-state brain activity of neurotypical adults, lower (or higher) reconstruction performance in fetuses or neonates can be interpreted as lower (or higher) similarity of their FC representations to the adult brain. Thus, future studies investigating the application of reconstruction degree in fetuses and neonates as a potential biomarker of diseases or a predictor of emotional/behavior outcome will be of interest. It needs to be pointed out, however, that the age dependency of reconstruction performance could be interpreted in other ways. It is possible that errors were introduced when projecting fMRI in individual brain space into the 40 wk standard brain template. In an attempt to reduce errors, we used a two-step approach for the registration: first, registering from the subject space to the age-matched brain space and second, from the age-matched brain space to the 40 wk template space. We accounted for the effect of brain size in the regression model. Thus, we believe our observation was minimally biased by projection errors; this, however, needs to be confirmed. Another possibility is that error originating from geometric reformatting accounted for the age dependency of reconstruction performance. We believe that this was unlikely, given that in our previous experiment with human adults (see Supplementary Figure 1 in *Kim et al., 2021a*), we showed that the error induced by geometric reformatting was minimal (*r* ~ 0.98 between the original cortical pattern and reformatted/recovered cortical pattern). Lastly, unlike the neonatal cohort, we found that there was a remaining negative correlation between age and pattern reconstruction performance, which was somewhat against our expectation, even after considering head size. Thus, we believe that caution is needed in interpreting the findings with the fetal cohort. To address this issue in the fetal cohort, fetal-specific model or analytical pipeline will be needed.

In adults *Kim et al., 2021a*, we hypothesized that our trained VAE model was able to extract generative factors of rsfMRI data as a form of latent variable. In the VAE, we speculated that learned generative factors, that is, latent variables, largely fall into two groups: one representing the common pattern of brain dynamics and the other shaping the unique brain patterns of individuals. Supporting this notion, in this study, we observed that a subset of latent variables of fetal–neonatal rsfMRI mapped to a set of large-scale FBNs, for example, visual, auditory, and sensorimotor, and default mode network (*Figure 7*, *Figure 7—figure supplements 1 and 2*), whereas another subset reflected the temporal properties of fetal–neonatal functional brain development (*Figures 5 and 6*, *Tables 1–3*). Additionally, we found that some subsets of latent variables (i.e., large-scale FBNs) varied with advancing PMA. For example, the brain network (IC6), which showed joint highlights in somatosensory and auditory brain regions, characterized by additional engagement of primary auditory network when neonatal brain matured. Similar pattern of the spatial involvement over aging was also observed in other brain networks (*Figure 8—figure supplement 1*, visual network, IC20, with engagement of somatosensory region). Spatial involving nature of neurodevelopment pattern during neonatal period, observed in our study, is largely in line with the trend in maturing brain system at older age that transit from 'segregation' into 'integration,' suggested by other studies (*Fair et al., 2009*; *Dosenbach et al., 2010*). Altogether, our results highlight that the VAE can identify spatial aspect of neurodevelopment during perinatal period, which are difficult to be achieved by conventional linear counterparts. Note that our observation does not preclude that subset of latent variables that were completely isolated; in fact, we suspect those subsets were likely highly overlapped as both tasks were done at the group level. Nonetheless, our findings suggest the delineation of generative factors of fetal–neonatal rsfMRI is plausible using the VAE model pretrained on adult rsfMRI without any further introduction of new fetal–neonatal fMRI data to the VAE training. Collectively, we believe that VAE may be used to create longitudinal maps of the functional brain connectome throughout the life span (i.e., a functional atlas from the fetal period to adulthood).

Given that the VAE model was trained to represent adult brain dynamics, it is unclear whether latent features of fetal–neonatal rsfMRI reflect 'real' aspects of the fetal–neonatal functional connectome (i.e., fundamental functional networks present during different developmental periods) or are driven by similarity/dissimilarity to the adult functional connectome (i.e., the evolution of patterns of brain

networks with aging). Our results suggest that both aspects are likely captured by latent features; some fundamental functional networks that remain largely consistent over neonatal period, reflecting the actual aspect of fetal–neonatal functional connectome. In contrast, some networks evolved over time (e.g., IC 6 in *Figure 8*); thus, some portions of their latent features may be driven by the dissimilarity/similarity to the adult network. Investigating this dichotomous aspect of VAE-derived latent representations will be another interesting topic as an application of the VAE in the fetal–neonatal neuroimaging field.

Central to this study was the improvement in spatial representation of fetal–neonatal fMRI cortical patterns via the VAE pretrained from resting-state brain patterns of adults. Once the fetal–neonatal rsfMRI data were projected onto the latent space defined by the VAE, we were able to model the temporal dynamics of latent representations at the group level using linear computational methods, for example, temporal ICA. To this end, we identified 30 independent latent bases and visualized each of them through the VAE decoder (*Figure 7*). Compared to FBNs defined by spatial ICA on the dHCP dataset (*Eyre et al., 2021*), brain networks reported in our study exhibited more dynamic patterns such as lateralized activation, bilateral activation, activation and deactivation across hemisphere, and interaction between brain networks (*Figure 7*, *Figure 7—figure supplements 1 and 2*). We believe the discrepancy in FBN patterns mainly originates from the methodological difference between temporal and spatial ICA. As shown in previous studies with adults (*Smith et al., 2012*), different from spatial ICA, temporal ICA outputs temporally independent functional modes having activation patterns along with deactivation patterns. Similarly, we also observed the opposition between sensory-related networks (#4: auditory vs. somatosensory; #16: visual vs. auditory, *Figure 7—figure supplement 1*). However, unlike the adult brain, we were not able to observe anticorrelated activity between cognitive networks and sensory networks, possibly suggesting such suppression of sensory functions via higher cognitive networks develops at a later neurodevelopmental phase. Another interesting observation was that the smoothing effect of the VAE was comparable to IC300 (similar effect to manual smoothing at the level of FWHM = 5 mm; *Figure 3*). Given the above, we believe the VAE model may be more suitable for investigating finer scale of brain networks than linear models. Perhaps, the VAE model with a greater number of latent variables (e.g., 512 or 1024 instead of 256 in the current VAE) can be utilized to find brain networks at finer scale. Altogether, we expect our proposed analysis scheme combining nonlinear spatial compression via VAE and temporal modeling via temporal ICA may be a critical addition to existing tools for mapping fetal–neonatal brain networks.

In general, we observed that FBNs identified in the DBI dataset were relatively less straightforward to interpret compared to the dHCP dataset, possibly due to many factors including, but not limited to, smaller scan size, shorter data length, different preprocessing steps such as ICA-FIX, or a combination of the above (see *Figure 7—figure supplements 1 and 2* for the whole set of FBNs from dHCP and DBI datasets, respectively). It is desirable to combine datasets collected from multiple centers to improve generalizability of findings and to boost detection power. Nevertheless, inconsistency of data acquisition parameter across different datasets, alike our study, acts as a technical challenge of inter-center rsfMRI analysis. Related, we found that FBNs defined by the linear models (IC50, IC200, and IC300) were more sensitive to inter-dataset difference, resulting in lesser pattern similarity between dHCP and DBI full-term groups, and greater similarity between pre- and full-term newborns within the dHCP dataset (*Figure 7D*). Opposite to linear models, we found that 30 FBNs derived by FBNs were more replicable across different datasets when their ages were matched. Collectively, we believe that the VAE model can be an analysis tool for big fMRI data collected throughout multiple centers. This is particularly crucial as it is relatively more difficult to collect rsfMRI data of fetus/neonate compared to the human adult.

While the VAE model was able to represent fetal–neonatal FC data, one limitation is that the model primarily focused on learning spatial representations of rsfMRI patterns and ignored the temporal dynamics of rsfMRI data. Functional MRI is not stationary (*Guan et al., 2020*), and our future work aims to address both spatial and temporal features of rsfMRI; currently, limitations on technical resources constrained our ability to simultaneously model both. One recent study done by *Qiang et al., 2021* tried to address this issue by introducing recurrent neural network to the VAE model. However, their model had to sacrifice the efficacy of their spatial representations by diminishing the geometrical structure of fMRI data (from 3D image +1D temporal to 1D image +1D temporal). Another recent study (*Brown et al., 2020*) was able to keep the 3D structure of fMRI data, but downsampled the

spatial resolution of input image grossly and reduced the computational layers for the sake of computational resources. Both models have achieved their desired temporal interpretability but at the cost of degraded spatial representations. Future studies with novel designs are needed to model the complex and multifaceted spatiotemporal dynamics of fMRI data. Another limitation of our study relates to the smoothing effect induced by the VAE which could render it insensitive to detecting fine-grained changes of early network development. As observed in *Figures 2 and 3*, the VAE model does exert some level of smoothing when mapping latent variables to the cortical space. This is, however, inevitable in most representational models (e.g., ICA in *Figures 2 and 3*). It should be noted too that the VAE outperforms linear models given the same degree of smoothing. We believe there are at least two plausible solutions to this limitation. First, we can increase the number of latent variables to 512 or 1024 from 256 (e.g., weaker smoothing effect in IC300 vs. IC50 in *Figure 3*), but at the risk of losing generalizability of representations (*Table 3*; worse cross-dataset age prediction performance in IC300 than IC50). Alternately, we can substitute the VAE with a generative adversarial network (GAN) as images generated by the latter is, by design, minimally smoothed (*Goodfellow et al., 2020*). Another limitation is that our VAE model currently does not capture information from the subcortex. This region plays a critical role in neural circuitry formation and including subcortical neural activity in future studies would greatly enhance our understanding of the developing functional connectome (*Fransson et al., 2009*; *Lordier et al., 2019*). Currently, we are unable to add the subcortical regions into the VAE model; doing so would require rebuilding the entire model architecture to include information such as subcortical projections into the neocortex. While there are validated adult deep learning models for how cortical hemispheric information is combined during convolution (such as hemispheric symmetry in the current implementation), to the best of our knowledge, limited data exist on how to match subcortical and cortical information. One possible way to circumvent this is to use the 3D volumetric rsfMRI activity as input to the deep learning models; however, as mentioned above, this could lead to significantly degraded spatial resolution. Alternatively, graph-based deep learning techniques could be explored. Other deep learning applications that could more effectively model information from subcortical regions should be further investigated. Lastly, the current VAE model was pretrained on adult rsfMRI data. Future studies incorporating newborn rsfMRI data would likely help emphasize features uniquely present in the developing brain.

In this study, we applied a novel deep generative model to a large (>500) fetal–neonatal rsfMRI dataset to extract nonlinear brain representations of healthy fetuses, preterm neonates, and full-term infants. In this study, we purposely utilized the VAE model pretrained using the human adult rsfMRI data. Our results suggest that the pretrained adult VAE model was applicable to fetal–neonatal rsfMRI data, but there is room for improvement. First, we expect that fine-tuning the VAE with fetal–neonatal rsfMRI will help the VAE to learn more meaningful network representations. Additionally, we can further optimize the hyperparameters of the VAE model. In the current model, the beta hyperparameter, which balances reconstruction performance and usefulness of learned latent variables, was set to 9, putting more weight on the latter term. This value was based on a previous study in adults, where the number was set to force latent variables to capture better representations of human adult rsfMRI data. Changing the setting of beta value to meet the specific goals of other studies will be of great important in future VAE applications for fetal–neonatal rsfMRI. Compared to linear methods, the VAE model generated improved representations of the brain's FC patterns and predict age more precisely. As a future work as an extension of this work, we plan to separate latent variables into two groups: age prediction accuracy at the group level vs. individual traits. Thus, once we can successfully isolate latent variables responsible for individual traits, we expect that subject identification task in the neonate group, which failed using the linear model (*Ciarrusta et al., 2021*), will be feasible using such pruned latent variables. A promising application of the VAE model may be for characterizing the developing brain's functional connectome. As the human brain develops rapidly over the fetal–neonatal periods, it is likely that the spatial pattern of FBNs will also evolve. For example, it has been suggested that a nascent default mode network (DMN) exists in fetuses (*Thomason et al., 2015*), while the DMN of neonates has been shown to closely resemble the human adult (*Doria et al., 2010*). However, due to the linear nature of conventional models such as ICA, it is difficult to capture the spatial evolution of FBNs over aging. For example, linear ICA can show changes in network strength but not changes in network spatial patterns with aging (see Figure 4 in *Eyre et al., 2021*). In contrast, the VAE model is nonlinear such that combining two latent variables in the latent space can yield continuous change

of network patterns in the cortical space (i.e., adding/subtracting age-related latent variable from DMN-related latent variable and projecting into the cortical space using the VAE decoder). Related to it, our findings showing a difference in age group-wise network patterns between latent vs. cortical spaces (*Figure 8—figure supplement 2*) suggested the possibility of the above mapping strategy that can track the neurodevelopmental growth of fetal–neonatal FBNs. Thus, the VAE model may provide important insights into fetal–neonatal neurodevelopmental growth trajectories. In-depth assessments of ICs in the latent space, each representing a large-scale brain network, can be a very interesting research topic. Among many, one feasible assessment can be a hierarchical grouping of ICs given their pattern similarity in latent representations and compare the settled hierarchy to human brain organization. Tracking transition from one IC to other ICs over time during resting state can be another assessment. New insights regarding the complex nonlinear brain development that occurs during the prenatal-neonatal continuum may allow us to identify early biomarkers for neurobehavioral vulnerability and link specific prenatal brain developmental processes associated with aberrant neural connectivity which may underlie prevalent child and adult neurological disorders.

## Materials and methods

We analyzed two large fMRI datasets (N = 727 scans), one dataset consisting of fetuses and neonates collected by our institute (n = 270), named as 'Developing Brain Institute' or DBI dataset, and one public dataset consisting of pre- and full-term neonates (n = 457), published by developing human connectome project, named as dHCP dataset (*Makropoulos et al., 2018*). Two fetal–neonatal datasets (DBI and dHCP), acquired by two independent centers, were utilized to test the utility and generalizability of the VAE model on representing fetal–neonatal rsfMRI data.

### DBI dataset: Fetus and full-term neonate

We analyzed 270 resting-state scans from 95 healthy fetuses and 160 full-term-born healthy infants. Scans were collected as a part of an ongoing longitudinal study examining prenatal–neonatal brain development at Children's National Hospital in Washington DC. Fetuses (49 females) from healthy pregnancies were scanned between 19.14 and 39.71 gestational weeks (mean and SD; 33.65 ± 4.01). PMA of full-term born infants at scan ranges between 37.71 and 47.43 wk (41.67 ± 1.84). Maternal exclusion criteria were psychiatric disorders, metabolic disorders, genetic disorders, complicated pregnancies, multiple pregnancies, alcohol, and tobacco use, maternal medications, and contraindications to MRI. All experiments were conducted under the regulations and guidelines approved by the Institutional Review Board (IRB) of Children's National (study ID: Pro00013618); written informed consent was obtained from each pregnant woman who participated in the study.

For fetal scans, structural and functional resting-state MR images were acquired using a 1.5 Tesla GE MRI scanner with 8-channel receiver coil. The structural MR images for the fetal brain were acquired using single-shot fast spin-echo T2-weighted images by following settings: TR = 1100 ms, TE = 160 ms, flip angle = 90°, and voxel size = 0.8 × 0.8 × 2 mm. Functional data were acquired using echo planar images (EPI) with TR = 3000 ms, TE = 60 ms, flip angle = 90°, field of view = 33 cm, and voxel size = 2.58 × 2.58 × 3 mm, and total scan volume = 144 (=7.2 min). The structural and functional MR images of full-term infant brain MRI studies were acquired using 3T GE scanner. T2-weighted fast spin echo MRI was obtained using the following parameters: TR = 2500 ms; TE = 64.49 ms, voxel size = 0.625 × 1 × 0.625 mm. The parameters of fMRI scans were set to TR = 2000 ms, TE = 35 ms, voxel size = 3.125 × 3.125 × 3 mm, flip angle = 60°, field of view = 100 mm, and total scan volume = 200–300 (=6.7–10 min).

Functional MR images were preprocessed as follows: slice time correction, discarding the first four volumes, de-spiking, bias-field correction, motion-correction, intensity scaling, detrending, bandpass filtering (0.009–0.08 Hz), and nuisance regression with motion parameters. We excluded volumes having excessive head motion (frame-wise motion >1 mm or rotational motion >1.5°). The number of volumes excluded by the above procedure was 32.4 ± 17.3 (equivalent to 64.7 ± 34.6 s) for the fetal fMRI. Detailed preprocessing steps for the fetal and neonatal MRI data can be found in *De Asis-Cruz et al., 2021a* and *Brown et al., 2020*, respectively. The preprocessing steps applied to the DBI dataset are summarized in *Supplementary file 1a*. Neonatal MRI scans with <4 min were excluded in the analysis. The analyzed data length of fetus and neonate group was 4–7 (mean ± SD

= 5.4 ± 0.9) and 4–8.9 (5.5 ± 0.8) min, respectively. Finally, preprocessed rsfMRI scans at the volumetric brain space were projected to the standard cortical space using HCP workbench command *-volume-to-surface-mapping*. The timeseries for each voxel was then normalized (zero mean and unitary variance). To the end, 21 neonatal scans (14 with short scan data <4 min and 7 with partial or full failure at volumetric registration) among 160 scans were excluded in the subsequent analysis.

## dHCP dataset: Preterm and full-term neonates

dHCP is an open science project approved by the UK National Research Ethics Authority (*Makropoulos et al., 2018*). FMRI data analyzed in the project can be downloaded at https://data.developingconnectome.org/. Among 464 subjects (261 F/203 M), we analyzed a total of 457 rsfMRI scans from 107 preterm and 302 full-term neonates. Their PMAs at scan range from 24.29 wk to 44.87 wk, and the mean and standard deviation of ages were 38.54 and 3.47 wk, respectively. It was reported that no babies in the preterm and full-term group had major brain injury. Demographical details regarding the dataset can be found elsewhere at *Russakovsky et al., 2015*. MR scans were acquired using a 3T Philips Achieva system (Philips Medical Systems). High-resolution T1- and T2-weighted anatomical imaging was acquired with followed parameter settings: T1-weighted image; spatial resolution = 0.8 mm isotropic, FOV = 145 × 122 × 100 mm, and TR = 4795 ms, T2-weighted image; spatial resolution = 0.8 mm isotropic, FOV = 145 × 145 × 108 mm, TR = 12,000 ms, and TE = 156 ms. Followed by anatomical scans, fMRI scans were acquired with acquisition specifications: multislice gradient-echo echo planar imaging (EPI) sequence, multiband factor = 9. TR = 392 ms, TE = 38 ms, flip angle = 34, spatial resolution = 2.15 mm isotropic, and total volume = 2300 (~15 min). The downloaded dHCP fMRI data was already preprocessed *Fitzgibbon et al., 2020* and registered into the volumetric brain template *Schuh et al., 2018*. The registration quality was assured by dHCP team (*Fitzgibbon et al., 2020*). As the primary reason for including the dHCP data was to prove that our findings were robust against specific preprocessing choices/data characteristics, we purposefully used the dHCP dataset without adding or modifying the preprocessing steps used by dHCP. We reviewed the quality summary provided by dHCP (file name: *sub-SUBID_ses-SESSID_funcqc.html*) and ensured that preprocessing/registration to the brain template was reasonable for each scan that we analyzed. To match the input format to the required input format of the VAE model, we wrapped volumetric fMRI data to the cortical space using HCP workbench command *-volume-to-surface-mapping*. Then, we applied additional preprocessing step consisting of voxel-wise detrending (regressing out a third-order polynomial function), bandpass filtering (from 0.01 to 0.1 Hz), and voxel-wise normalization (zero mean and unitary variance). To avoid possible distortion from the filtering step, we rejected first and last 150 volumes from the analysis. Lastly, to prevent the possible confounding effects from motion artifact, we selected 1400 volumes (~10 min) having the least frame-wise displacement degree, per subject. Note that the concatenation of discontinuous rs-fcMRI data does not significantly distort FC *Fair et al., 2007*.

## Pretrained variational autoencoder using adult rsfMRI

In our previous work *Kim et al., 2021a*, we designed a variation of β-VAE *Higgins, 2017*; *Kingma and Welling, 2013* to learn representations of rsfMRI patterns of adults. Briefly, our proposed VAE model consisted of an encoder and a decoder. After reformatting the fMRI pattern at the cortical space to the carefully designed 2-D regular grid space (*Figure 1A*), an encoder compressed an fMRI map (no. of vertices = 38,864) to a probabilistic distribution of 256 latent variables, whereas a decoder reconstructed the fMRI patterns given the sampled latent variables (*Figure 1B and C*). The encoder and decoder consisted of five nonlinear convolutional layers and five de-convolutional layers, respectively. The overarching goal of the VAE model was to reconstruct input at best under the constraint forcing the distribution of every latent variable to be close to be i.i.d. Specifically, given the training dataset $X$, we optimized the encoding parameter $\phi$ and the decoder parameter $\theta$ to minimize the loss function defined as

$$L\left(\phi, \theta | x\right) = \left\| x - x^{'} \right\|_{2}^{2} + \beta \cdot D_{KL}\left[ N\left(\mu_{z}, \sigma_{z}\right) \parallel N\left(0, I\right)\right]$$

where $x$ and $x^{'}$ are the input image and the reconstructed version of input image, $\beta$ is a hyperparameter balancing two terms in the loss function, $D_{KL}$ is the Kullback–Leibler (K-L) divergence

between the posterior distribution $N(\mu_z, \sigma_z)$ and prior distribution $N(0, I)$, and $\mu_z$ and $\sigma_z$ are mean and standard deviation of estimated latent variables $z$ from $x$. Upon validation dataset consisting of 50 young adults rsfMRI data from Human Connectome Project (*Van Essen et al., 2013*), we optimized the hyperparameters of VAE model: total number of layers = 12, learning rate = $10^{-4}$, batch size = 128, and $\beta$ = 9. After being trained with another large HCP rsfMRI data consisting of 100 young adults, the trained VAE model has learned to represent patterns of cortical activity and network patterns using latent variables. We refer to our original paper *Kim et al., 2021a* for the detailed description of geometric reformatting and the model architecture of the VAE. As our goal of this study was to utilize brain representations learnt from adults for better understanding of fetal–neonatal neuroimaging data, we froze the model parameters of the VAE model, $\phi$ and $\theta$, and froze the VAE model hyperparameters. Therefore, the VAE model used in this study was identical to the VAE model reported in our original paper (*Kim et al., 2021a*). In this study, the dHCP and DBI datasets were only used for testing the pretrained VAE model. Latent variables estimated from cortical patterns of the dHCP and DBI datasets were used as features of age prediction experiment or networking mapping (*Figure 1D*).

## Reconstructing fetal–neonatal rsfMRI patterns using the pretrained VAE

We evaluated how well the VAE model that was pretrained to represent adult rsfMRI delineated the fetal–neonatal brain activity that was unseen during the training. We compressed the spontaneous cortical patterns of the dHCP and DBI datasets using the pretrained VAE encoder and restored them to the cortical space by feeding sampled latent variable to the VAE decoder. The reconstruction degree of each time point was defined as the Pearson correlation coefficient between the original- and reconstructed cortical patterns. Averaging reconstruction performance over timeseries or subjects was done after converting $r$-value into Fisher's $z$-score. To evaluate the smoothing effect of VAE and linear models, we manually smoothed the original rsfMRI data using HCP workbench code *-cifti-smoothing*. The smoothing size was defined as full-half-maximum-width (FWHM) and varied from 1 to 10 mm at 1 mm intervals.

## Predicting gestational age and postmenstrual age of fetus, preterm neonates, and full-term neonates

To evaluate the age prediction power of latent variables defined by the VAE, we built an age prediction model and validated its prediction accuracy using tenfold cross-validation scheme. For dHCP dataset, we utilized FC profile at the latent space as individual-wise features (*Figure 5A*). Specifically, cortical pattern of each time point was encoded as timeseries of latent variables using the VAE encoder. Next, we measured FC profile (no. of features = 256 × 256/2) of each subject by measuring covariance between every pair of timeseries of latent variables. After keeping 10% of dataset as a testing group, we further divided the remainder 90% of dataset into training (=76.5%) and validation set (=13.5%). Within the training dataset, we applied the feature selection procedure by correlating the age and each feature. We thresholded features having FDR-corrected p<0.05. Based on the selected feature set, we built a linear regression support vector model (RSVM) using MATLAB function *fitrsvm.m*. In comparison to other age prediction studies (*Gong et al., 2021*), our regression model also tended to output biased error toward extremity of age distribution, called as a prediction bias. To eliminate such prediction bias, we further added an adjusting step of predicted age using the validation dataset. Specifically, we predicted the age of validation dataset using the trained RSVM model and built a linear regression model between predicted- and actual age of validation set. After calculating the optimal regression fit, we compensated the prediction age by the trained RSVM model. Finally, the prediction performance of designed model was tested using the unseen testing dataset, repeating the whole procedure for 10 folds independently. The whole tenfold cross-validated age prediction task was independently repeated for 100 times, each with randomly grouping scans into test/train/validation sets. The final reconstruction performance was reported by averaging over 100 trials. The sample size of final test/train/validate set was 314/55/40.

The prediction model for the DBI dataset was nearly identical to the above prediction model in dHCP dataset, except that, in the feature selection step, we took the feature positively correlated to the age and summed the significant features. We modified this step to make the model more robust against possible bias due to different recording parameters (e.g., spatial resolution) in the fetal and

neonatal groups. Moreover, during reconstruction, we observed that the association between age and reconstruction performance trended in opposite directions in the neonates and fetuses. Thus, to ensure that the age prediction model was applicable in both datasets, we modified the feature selection step. As the dimension of the feature set became 1, we employed a linear regression model using MATLAB function *fitlm.m*. Note that, except for the feature selection step and regression model type used, all other steps were identical to those used in the dHCP dataset. We also performed age prediction on neonatal and fetal datasets of the DBI separately. For this, we modified the prediction model in two ways. Since the number of samples was limited compared to the combined fetus/neonate model, we skipped the step adjusting for age prediction bias; thus, 90/10% of data was used for the training/testing set, respectively. Another modification was adjusting the feature selection threshold from FDR-corrected $p<0.05$ to uncorrected $p<0.01$. The whole tenfold cross-validated age prediction task was independently repeated for 100 times, each with randomly grouping scans into test/train/validation sets. The final reconstruction performance was reported by averaging over 100 trials.

Finally, we built an age prediction model generalizable across different datasets, dHCP and DBI (*Figure 6A*). We took the whole dHCP dataset as training/validation set for the model and tested the prediction performance on the DBI dataset. We separated the dHCP dataset into testing set (=85%) and validation set (=15%). The age prediction algorithm was identical to one employed in the DBI dataset, as two datasets had different spatial and temporal resolution (e.g., 10 min for dHCP vs. 4–6 min for DBI). The whole age prediction task was independently repeated for 100 times, each with randomly selected 50% of dHCP dataset (n = 205 scans). The final reconstruction performance was reported by averaging over 100 trials.

The prediction performance was evaluated using four different metrics, RMSE, MAE, squared correlation coefficient per fold ($r^2$), and correlation coefficient between actual age and predicted age at the whole dataset level (correlation).

## Mapping of fetal–neonatal brain networks using pretrained VAE

We examined the utility of latent variables defined by the VAE as a mapping tool of fetal–neonatal FBNs (*Figure 7A*). First, we estimated the timeseries of latent variables by feeding each fMRI time point to the VAE encoder and concatenated the timeseries of latent variables across subjects for different age groups; preterm baby (<37 wk) vs. full-term neonates (≥37 wk) for dHCP dataset and fetuses vs. full-term neonates from the DBI dataset. We redefined the latent space by applying temporal ICA to concatenated latent variables. The temporal ICA was applied using the Infomax ICA algorithm, which was implemented by MATLAB toolbox EEGLAB (*Delorme and Makeig, 2004*). Inspired by the recent ICA study investigating neonatal FBNs (*Eyre et al., 2021*), we also set the number of ICs to 30 heuristically. Once 30 independent latent bases were identified, cortical mapping of each was done as followings: (1) we multiplied random scaling factors (n = 1000) to the latent basis, (2) reconstructed 1000 cortical patterns of scaled latent basis using the VAE decoder, and (3) calculated covariance between random scaling factors and reconstructed fMRI activity, per cortical location. The estimated covariance scale of each cortical location was considered as the activation level of the location and the sign (+/−) of covariance was defined as activation (+) or deactivation (−) state. We fixed the sign of each cortical map as the region having the strongest covariance scale being activation (+) state. Note that as the VAE decoder is highly nonlinear, our proposed mapping method from latent space to cortical space did not guarantee to reflect the true cortical meaning of independent latent bases, but the results in our study empirically suggested that our proposed method was a good approximation mapping strategy. Lastly, the cortical map of each basis function was heuristically thresholded at the 15% of the maximal absolute value of each map, for the better interpretation of results.

We further examined group-wise reproducibility of estimated independent latent bases across age groups and/or different datasets. Similarity between latent bases was measured by estimating the Pearson correlation between every latent basis from one group and every basis from another group. As the sign of independent latent bases was somewhat arbitrarily determined in ICA algorithm, we one-to-one paired latent bases across age groups based on their absolute correlation coefficient values. We took the trend of paired similarity over increasing IC numbers as the inter-group reproducibility of brain network. For the sake of reproducibility, the whole analysis was independently repeated for 100 times, each with different initialization seeds for ICA. The final reproducibility trend was reported by averaging over 100 trials.

## Comparison with linear representation methods

To access the utility of the VAE model, we employed conventional linear representation methods, cortical parcellations, and group ICA maps. For the sake of fair comparison between representation methods, we utilized multi-modal cortical parcellation (*Glasser et al., 2016* and group ICA maps *Smith et al., 2013*), both were derived by the HCP dataset. In the employed cortical parcellation, there were 360 areas (180 per hemisphere), and we compressed fMRI data at the cortical space into the linear latent space (no. of dimension = 360 cortical parcels). We also utilized group ICA results provided by HCP, which is freely available at https://db.humanconnectome.org/. Briefly, group ICA maps were estimated based on the rsfMRI of 820 subjects in the HCP and the complete algorithm of group ICA can be found at *Van Essen et al., 2013*. This group ICA provides different number of latent variables (or no. of IC), 15, 25, 50, 100, 200, and 300. Among them, we utilized IC maps consisting of 50, 100, 200, and 300 ICs (named as IC50, IC100, IC200, and IC300). To match the number of latent variables, ICA maps with 256 components (named as melodic ICA) were estimated by applying melodic ICA *Beckmann and Smith, 2004* to HCP adult data that were used for training the VAE model. Similar to the VAE model, we compressed cortical activities into IC activities using the pseudoinverse of IC maps and restored them using the IC maps. In the regime of linear analytical models, complex rsfMRI data are considered as a linear combination of spatial representations. Specifically, fMRI activity ($X$, dimension: 59,412 × 1400 for dHCP dataset) was compressed by multiplying with the linear IC map ($A$, matrix dimension: 59,412 × 300 for 300 ICs), such as $Y = A^\dagger X$, where $Y$ (size: 300 × 1400) is the timeseries of IC and $A^\dagger$ is the pseudoinverse of $A$. Then, the reconstructed fMRI activity $\widetilde{X}$ was calculated as $\widetilde{X} = AY$. Same as the VAE model, the reconstruction performance was measured by correlating between original- and reconstructed cortical patterns. Linear latent variables defined by different IC maps were utilized for age prediction task as well.

## Acknowledgements

We thank the participants of this study. This study was funded by grant R01 HL116585-01 from the National Heart, Lung, and Blood Institute, National Institutes of Health and grant MOP-81116 from the Canadian Institute of Health Research.

## Additional information

### Funding

| Funder | Grant reference number | Author |
| --- | --- | --- |
| National Heart, Lung, and Blood Institute | R01 HL116585-01 | Catherine Limperopoulos |
| Canadian Institutes of Health Research | MOP-81116 | Catherine Limperopoulos |

The funders had no role in study design, data collection and interpretation, or the decision to submit the work for publication.

### Author contributions

Jung-Hoon Kim, Conceptualization, Resources, Software, Formal analysis, Validation, Investigation, Visualization, Methodology, Writing – original draft, Project administration, Writing – review and editing; Josepheen De Asis-Cruz, Data curation, Supervision, Funding acquisition, Investigation, Methodology, Writing – review and editing; Dhineshvikram Krishnamurthy, Resources, Software, Methodology; Catherine Limperopoulos, Conceptualization, Resources, Data curation, Supervision, Funding acquisition, Validation, Investigation, Methodology, Writing – review and editing

### Author ORCIDs

Jung-Hoon Kim http://orcid.org/0000-0003-3032-8827
Catherine Limperopoulos http://orcid.org/0000-0003-1735-0069

### Ethics

Human subjects: All experiments were conducted under the regulations and guidelines approved by the Institutional Review Board (IRB) of Children's National (Study ID: Pro00013618); written informed consent was obtained from each pregnant woman who participated in the study.

### Decision letter and Author response

Decision letter https://doi.org/10.7554/eLife.80878.sa1
Author response https://doi.org/10.7554/eLife.80878.sa2

---

## Additional files

### Supplementary files

• MDAR checklist

• Supplementary file 1. Supplementary tables. (**a**) Preprocessing steps for the DBI and dHCP datasets. (**b**) Age prediction performance in separate age groups of DBI dataset using different latent representations.

### Data availability

Data from the Children's National cohort (or DBI dataset) are accessible here: https://doi.org/10.5061/dryad.cvdncjt6n. The Developing Human Connectome Project dataset (dHCP dataset) are here: http://www.developingconnectome.org. The source code, model and documentation for the VAE described in this paper are publicly available at https://github.com/libilab/rsfMRI-VAE (*Kim et al., 2021b*).

The following dataset was generated:

| Author(s) | Year | Dataset title | Dataset URL | Database and Identifier |
|---|---|---|---|---|
| Kim J, De Asis-Cruz J, Krishnamurthy D, Limperopoulos C | 2023 | Towards A More Informative Representation of the Fetal-Neonatal Brain Connectome using Variational Autoencoder | https://dx.doi.org/10.5061/dryad.cvdncjt6n | Dryad Digital Repository, 10.5061/dryad.cvdncjt6n |

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
