## [Editor Report]

Presenting important findings, this study describes the development of the functional brain connectome in human fetuses and neonates through the application of a novel deep learning approach: adult trained variational autoencoder. The methodology, analyses, and evidence provided are convincing and pave the way for future studies on non-linear models of brain network maturation. This work is of potential neuroscientific and methodological interest to researchers studying functional resting-state networks and brain development, as well as to deep learning scientists.

---

## [Decision Letter]

**Decision letter after peer review:**

Thank you for submitting your article "Towards A More Informative Representation of the Fetal-Neonatal Brain Connectome using Variational Autoencoder" for consideration by *eLife*. Your article has been reviewed by 3 peer reviewers, and the evaluation has been overseen by a Reviewing Editor and Floris de Lange as the Senior Editor. The following individual involved in the review of your submission have agreed to reveal their identity: Yong He (Reviewer #1); Andrea Gondova (Reviewer #2).

As detailed in their individual reviews, all three reviewers acknowledged the potential insights from this study and provided several valuable comments on it. Among them, we identified essential revisions that should be carefully considered so that the study can be suitable for publication in *eLife*.

1. Importantly, the current manuscript indicates that this approach captures non-linear patterns of development in fetuses, neonates, and infants, but there appears to be no analysis or results to support this claim. Without this aspect, the study lacks multidisciplinary interest. Furthermore, the biological and developmental significance of the brain states and networks revealed by the VAE approach with respect to functional systems needs to be better explored and compared to other methods. Besides, the current approach did not include subcortical structures that are known to play a key role in the development of functional networks (e.g. transient developmental layers like the subplate, the thalami), which requires further analysis or discussion.

2. The potential of the whole approach to describe the emergence and development of functional connectivity needs to be tempered as the use of a VAE model trained on adult data is based on the strong assumption that connectome features are rather similar/stable across ages. Therefore, it cannot reveal structures/networks that would be strongly different (e.g. present at early stages but not in the adult brain). The smoothing effects inherent in VAE also make the method relatively insensitive to the fine granularity of developing networks. In addition, volume registration may be less effective than surface alignment given the intense growth and increased folding over this age range.

3. The performance of the reconstruction might be driven by several factors other than functional information and maturation, most of which are also age-dependent: brain size and folding (in relation to registration issue), head motion (especially within the womb), acquisition settings (in fetuses vs. neonates), scan duration, processing pipeline, etc. In addition, differences in preprocessing between the two datasets (dHCP vs. DBI) could explain part of the differences in reconstruction, prediction, and mapping findings. It is important to disentangle the effects of potential confounding factors from the developmental mechanisms to consolidate the current findings and interpretation.

*Reviewer #1 (Recommendations for the authors):*

There are some vague expressions and potential methodological biases throughout the manuscript, which should be further addressed prior to publication.

1. The authors emphasized representing the non-linear maturation of brain functional maps from fetal to neonatal stage by their methods throughout the manuscript. But they failed to interpret the contribution of the proposed approach for capturing this non-linear development. Their model is pre-trained by adult data. The non-linear effect of variational autoencoder refers to the non-linear combination of representing fMRI signals between brain vertexes at the individual level. The age prediction is performed using the SVM model. None of these is directly related to non-linear development. Which result and analysis support this hypothesis?

2. Some potential biases in the comparison with linear ICA methods in age prediction analysis should be further evaluated. First, the number of latent variables and the number of ICA components should be kept the same during the comparison. Second, the effect of individual functional mapping on the baby brain should be taken into account.

3. The advantages of the VAE approach in generating brain functional networks at the system level are not exhibited. Do their methods find more meaningful functional systems than previous studies? This should be well evaluated.

*Reviewer #2 (Recommendations for the authors):*

The paper is a very interesting application of the VAE method to the early brain. This represents an important amount of work that went into the analysis and presentation of the results. The work is easy to navigate and clearly written (some small problems are detailed below), although the supplementary information cited within the text was not made available. The figures are of good quality and clear to interpret.

• Both data sets have a fairly large range of PMA at scan (in DBI for foetuses from 20w GA). Fast and large-scale changes are taking place within the developing brain in this period which might affect the data processing pipeline at all stages. Please include more details if any quality checks were performed. E.g.:

– Any rsfMRI QC for dHCP data;

– For dHCP, babies were included independently of their radiological scores -> this might have a significant impact on the reconstructed functional data;

– How good was the projection to cortical space? (Authors use 40w template, there is a large temporal distance for some babies – there might be errors distributed non-randomly and based on ages);

– Geometric reformatting – errors with inflation, 3D->2D?, if, are those distributed randomly (unlikely);

– During dHCP data description authors say 'The additional step of voxel-wise detrending, bandpass filtering and voxel-vise normalization' after volume-to-surface mapping. Were these steps applied to DBI set as well?

– Overall, more detail on the processing pipeline(s) would be useful with details on possible differences between datasets and focus on limitations that stem from it on the interpretation of results.

• It would be useful to perform additional data analysis between the two datasets before there are forwarded to the VAE – is there a significant difference between datasets that are due to the pre-processing? Could these explain differences in reconstruction, predictive, and mapping differences?

• It would be very useful to focus on the performance of the VAE compared to adults given the method was developed and validated in adults and most of the novelty comes from its re-application to infants. Please, detail how the reconstruction compares in DBI, dHCP vs adults. It would be good to think about other interpretations for this difference on top of the neurodevelopmental stage (approximated by PMA at scan)- brain size, projection to the template, geometric reformatting to 192x192 array.

• Correlation of reconstruction error with PMA at scan (Figure 3).

– Figure 3B: It would be good to incorporate more detail on the 3 outlier points in DBI dataset – why does it fail so much – do these kids happen to have low quality recording? Or maybe injury that makes the model fail? Understanding why it fails might give informative cues on what the model is actually trying to represent.

– Also, the 3 outlier points in DBI dataset are in the 'normal' cloud in dHCP dataset, please discuss why the reconstruction might be so different between DBI and dHCP datasets

– Please show reconstruction with age for the foetal data. Discuss more why this negative correlation – it can be smoothing, differences in processing, but also movement etc. – in any case, these points make us expect the relationship to be weaker but not to be strong and in the reverse direction, comment more on the foetal processing pipeline

– You are taking head motion into account between sessions – however there might be a relationship between motion and age => it would be good to re-run the same analysis as in reconstruction vs age but with motion

– It would be interesting to see the same comparison for the ICA – e.g. what are the correlations for the second best method IC300? Is that capturing more age/individual signature/noise? If the difference is significant would be a supporting argument for improvement using VAE.

• In the age prediction methods, please indicate final feature set sizes after feature selection to give an idea of what proportion of the latent space might be representing age information, what is the interpretation for the features not associated with age? (if individual signature, could you look into the predictive power of the latent vector to identify the individual in the future?)

• Regarding comments on summing the significantly correlated features (again specify how many) 'to make model more robust against different recording specifications' and then L203: 'age prediction in the dHCP dataset was better than in the DBI dataset, this is likely due to the shorter scan duration' => this makes if for an unfair comparison, in the summing – the latent space is collapsed into one feature which will necessarily remove information, additionally, the reconstruction of the DBI data (specifically foetuses) showed strange relationship with age. Moreover, the processing was different (with its own difficulties for foetuses). There are more limitations other than shorter scan durations involved, please discuss these

• Please detail how the prediction fares on neonates vs foetuses in light of reconstruction with age relationships being very different

• L278 makes it sound the evaluation was made at the age group level separately, this is not the case but would be useful. Overall, it would be beneficial to first restrict the analysis to a narrow age range, for example, the term-equivalent age with the homogeneous data acquisition settings and processing steps to validate the application to dissociate whether the results really do capture the variation within the functional connectome rather than all the other potential confounders.

• L250: does it make sense the author found precursor of the default mode network in DBI and not in dHCP given the DBI subjects are generally younger than dHCP and thus further from the cited young adults rsfMRI studies?

• L254-256: shorter data length is only one of the potential interpretations, another alternative (among others) might be that the maps are not reliable and mostly representing the noise in the data. In the discussion it would be good to go deeper into differences between the networks between the datasets and age groups and whether these make sense in light of the expectations. Can we expect to find the same FBNs across the lifespan? Are the ones identified early on less complex/ordered, or involving primary networks rather than associative ones?

• L263: 'least pattern similarity was observed between preterm neonates and fetuses in the DBI dataset'. This is interesting – is there difference between PMA at scan between these two groups? In light of this observation, does it make sense to include preterm subjects in the age prediction models – and other interpretations (rather than keeping them separate)? Similarly, does it make sense to include fetuses given differences in acquisition settings and post-processing? This brings back the earlier suggestions to initially keep the whole analysis to a narrow range of heterogeneous data to validate the method && increase the confidence that the extracted maps are 'real' rather than artefact of processing.

[Editors' note: further revisions were suggested prior to acceptance, as described below.]

Thank you for resubmitting your work entitled "Towards A More Informative Representation of the Fetal-Neonatal Brain Connectome using Variational Autoencoder" for further consideration by *eLife*. Your revised article has been evaluated by Floris de Lange (Senior Editor) and a Reviewing Editor.

The manuscript has been improved but there are some remaining issues that need to be addressed, as outlined below:

We all acknowledge that your revisions have significantly improved the methodological scope of the study, but the methodology described is not new and the neurodevelopmental insights still seem insufficient given the scope of *eLife*. The potential for a future study is mentioned, but it is such beginnings of evidence that we would like to read in this manuscript. This being said, we would like to offer you a new revision if you believe you'll be able to put more emphasis on some neurodevelopmental findings. Here is a suggestion of analysis that would seem relevant to us.

As we expect that distinct functional networks might evolve differently with development, you might consider different groups of neonates according to age (not fetuses for which results are less clear), highlight between-groups differences in the estimated weights of latent features derived within the autoencoder, and provide a characterization of the latent features showing the highest contribution differences in relation to age (it would be helpful to somehow relate these features to functional networks that can be visualized in terms of brain spatial topography). In our opinion, such analysis would provide some insights on the functional features and networks that show dramatic changes during this developmental period, although such features are derived from the adult brain.

If you are not convinced by this suggestion, please propose another analysis that you think would be more relevant to show some insights on the developing functional connectivity from your methodology.

It would also be very welcome to emphasize that a future study should provide a new training procedure in babies and integrate sub-cortical structures, in order to highlight features potentially present in the developing brain but not in the adult brain.

Finally, we would like to stress that multiple rounds of revisions are not usual in *eLife*, thus we hope that you will be able to provide a new manuscript fitting with our expectations. Otherwise, we would advise you to submit your work to another journal with more focus on methods.

*Reviewer #2 (Recommendations for the authors):*

We would like to commend authors for the amount of additional work performed in response to the initial round of reviews. Despite the incorporation of additional analyses and methodological details, the manuscript remained clear and informative, with the same high quality of presentation, analysis, and figures. Additionally, we were pleased to hear about the plans for future work in this interesting area (e.g. further evaluation latent representations' content and their usefulness for the individual identification).

Few additional comments and reflections incited by the revised manuscript and authors' replies are detailed below with hope they might be helpful for their future work.

1. Adult to neonate: how to interpret the resulting networks?

On Page 5 of the response to comments file, the authors bring up an important point regarding the adult lense through which the community tends to analyse early connectomes. As mentioned by the authors, this issue is shared by the majority of papers within this space and we believe it would be useful to discuss this topic explicitly in regards to the limitations it creates for the interpretation of results. With such a point of view, do the authors envisage the latent representations derived from the infant data to be 'real' features of the functional connectome at a given time point ('in-themselves') or rather 'dissimilarities/distances' to the adult network ('in-relation-to' the training data)?

In both of these cases, the latent representations will be practically useful, and outcompete non-linear models, with a high potential for improving the age, individual, and possibly atypical development prediction. However, the interpretation shift – VAE deriving features relevant to age prediction (supported by improved age prediction) to features relevant to brain development (suggested on Page 16 of the response to comments) – might be difficult to make until the deeper understanding of the content of derived latent representations is reached. Comparing the 30 networks derived from the models finetuned to different age ranges (as suggested by the authors) could be very interesting to shed more light on the developmental dynamism and this question. Until then, more caution might be warranted in the current setting when incorporating concepts like 'longitudinal' investigations (L81) or 'neonatal brain dynamics (L109).

On a slightly unrelated note, the authors then argue that 'utilizing adult rs-fcMRI at the level of the analytical model is more desirable than at the interpretation level, as there may be information from rs-fcMRI in adults that can more objectively guide the way we understand representations of neonatal rs-fcMRI'. Is this really the case? Our intuition would be that the bias introduced in one step of a pipeline is likely to be propagated in the consecutive analyses. Thus, it is difficult to see why introduction of the adult reference earlier in the research pipeline, at the level of the model, would be more desirable rather than for example on the contrary more obscuring?

[Additionally, we wonder if it would be useful to have an age prediction baseline which does not use the functional inputs at all? Would comparing the age prediction performance using basic parameters (for example head size, GA at birth to predict PMA at scan) to a model trained on VAE inputs give some additional information about the VAE representational ability vs the cohort composition effects?].

2. Radiological score and reconstruction performance

We are hesitant regarding the presentation of the lower reconstruction performance in neonates with the radiological score=5 (Page 16 of the response to comments file, and then within the manuscript L452). The radiology score of 5 describes an 'incidental finding with possible / likely significance for both clinical and imaging analysis'. Thus, it is expected (and reassuring) that the authors observed reconstruction differences between score groups as changes to the typical brain might additionally disrupt the functional networks independently of the developmental stage of the infant, i.e. make them more unlike the adults. However, these subjects will also probably suffer from a potential decreased quality of the image processing/analysis, for example due to lesions, which could confound these results. Thus, while we think it would be very interesting to further investigate the latent representations of the infants in regards to the brain injury (for example prediction model that classifies infants into subgroups based on the radiological score within the infant group of similar age), it might be difficult to understand how to dissociate the effects of age, injury, and processing quality on reconstruction and predictive performance in this group without additional discussion and quality checks (for example does the data and registration quality differ between different radiology score groups?). It might be more straightforward to focus on the observation of significant results in 'typical' groups (radiological score-1 and/or 2) as a support for potential of VAE to provide a surrogate measure of the distance of infant-to-adult activity pattern and clarify further the reasoning/caveats regarding the results in the score=5 group.

3. Foetal results

Moreover, we remain hesitant regarding the interpretation of foetal results and therefore the use of 'foetal-neonatal' outlook throughout the manuscript.

As mentioned on the L197: 'Once brain size [in foetuses] was accounted for, the negative correlation between age and reconstruction performance was reduced'. This suggests that some, but not all of this unexpected behaviour was accounted for by the head size and the ensuing ballooning effect. If we assume that the young foetuses should not be more similar to the adult cohort than the older ones in terms of the reconstruction performance, is it correct to expect additional problems remaining within the reconstruction of the foetal data?

Additionally, the statement on L40 that '[VAE led to] improved prenatal-neonatal brain maturational patterns and more accurate and stable age prediction compared to linear models' is not entirely accurate given the results presented on L280 which reports that it was cortical parcels that showed the best performance in the foetal group.

Thus, it might be difficult to conclude on L361 that 'This finding may suggest that neurodevelopment of preterm neonates and age-matched in-utero fetuses likely differ given the early extrauterine exposure in infants born prematurely, above and beyond the difference in acquisition settings and preprocessing steps between dHCP and DBI datasets.' Although we agree with the statement in itself, because of the remaining above-mentioned issues it might be difficult to use current findings as a support.

Given the remaining questions regarding the reliability of the foetal results, it is possible that this population (due to very different acquisition requirements, processing problems, large developmental distance) might benefit even more than the other groups from a foetal-specific model/pipeline (whose creation was suggested by the authors). Until then, more caution might be helpful when promoting the foetal aspects of the study.

*Reviewer #3 (Recommendations for the authors):*

The authors have provided very comprehensive responses to the comments and have largely addressed things appropriately by changing the wording in the manuscript itself.

Although they do a good and compelling job showing that the VAE method is seemingly outperforming other methods (in this case ICA), I have to confess I still struggle somewhat with the insight that can be drawn from this work – apart from the "potential" of the method to study early brain development. Most prominently, the "non-linear" aspects of the method are heavily emphasised in the manuscript – which for characterising brain development would of course be a major benefit which would be unique to this method. However, as it stands, the authors concede that developmental trajectories have not been explored and it could be done in later work. Rather than suggesting there is "potential for this" however, I suspect a reader would want evidence that this method can definitely provide this kind of novel insight to conclude whether it is truly characterising biological effects and/or is worthwhile?

This issue about demonstrating whether this method is truly a specific tool that can provide new insights about brain development are also relevant for 2 other unresolved issues – which would be key to differentiate if this is just a paper showing that a method can be used on fetal/neonatal data or if it is a method that is specifically worthwhile using on these populations. These are again are conceded by the authors but again readers I suspect, may want answered. However, I fully appreciate that these cannot be easily resolved. These issues revolve around the use of the adult training data which means: that (i) the subcortical structures cannot be included – these structures one might argue are even more important in fetuses than the cortex? and (ii) the distinction between "model generation (VAE) and interpretation (ICA)" becomes important. The authors here argue that generation is more desirable than interpretation – although I would personally argue the other way, as interpretation means that anything can be identified but can then be explored (as opposed to generation, where it would not be identified in the first place?). Age effects using ICA networks for example can be easily studied using a method like dual regression. It would be nice to see what would happen if a neonatal data set for example was used for training – but failing that I think it would benefit the reader if this specific issue is explicitly discussed in the same way as in their response?

---

## [Author Response]

As detailed in their individual reviews, all three reviewers acknowledged the potential insights from this study and provided several valuable comments on it. Among them, we identified essential revisions that should be carefully considered so that the study can be suitable for publication in eLife.1. Importantly, the current manuscript indicates that this approach captures non-linear patterns of development in fetuses, neonates, and infants, but there appears to be no analysis or results to support this claim. Without this aspect, the study lacks multidisciplinary interest. Furthermore, the biological and developmental significance of the brain states and networks revealed by the VAE approach with respect to functional systems needs to be better explored and compared to other methods. Besides, the current approach did not include subcortical structures that are known to play a key role in the development of functional networks (e.g. transient developmental layers like the subplate, the thalami), which requires further analysis or discussion.

Regarding the comment “… the current manuscript indicates that this approach captures non-linear patterns of development in fetuses, neonates, and infants, but there appears to be no analysis or results to support this claim. Without this aspect, the study lacks multidisciplinary interest.”

While the VAE captured features of brain functional connectivity that improved representations of resting state networks, predicted more precisely GA/PMA, and yielded outputs that were more generalizable compared to linear models, we did not, as the reviewers rightfully pointed out, provide direct evidence that VAE captured the non-linear components of fetal-newborn brain development. Guided by the reviewers’ critical feedback, we revised the manuscript to instead highlight VAE’s ability to capture elements of FC that are not picked up by linear methods (i.e., rather than VAE capturing non-linear features of brain development). Next, we discussed how the VAE model’s ability to extract non-linear patterns in data may help reveal non-linear patterns of brain maturation. Although the current study did not directly relate VAE outputs to non-linear neurodevelopmental trajectories, we think that it provides a necessary foundation and a sound rationale for subsequent investigations aiming to do so. Thus, while lacking this element, we believe the manuscript still maintains high scientific standards and maintains its multidisciplinary appeal. We hope, that with the extensive revisions, the editor and reviewers concur.

Here in our point-by-point responses, we share preliminary findings that begin to address our contention that the VAE model could meaningfully model non-linear brain growth. This result would not be included in the main manuscript, rather, it would be included in a follow-up paper where the proposed method’s capacity to capture non-linear features of development could be more carefully and exhaustively explored.

Regarding the comment “… the biological and developmental significance of the brain states and networks revealed by the VAE approach with respect to functional systems needs to be better explored and compared to other methods.”

As suggested, we expanded our discussion on the biological significance of the brain networks revealed by the VAE model compared to linear models. Newborn networks discovered with the VAE model using the dHCP dataset were reproduced in the age-matched DBI dataset (revised Figure 7.D; see page 37 line 823-832 for the details of methods). This similarity suggests that the model is likely homing in on features related to brain physiology instead of variations in acquisition, scan setting, and preprocessing steps. If the VAE model was more sensitive to the latter factors, networks derived from the two age-matched, independent datasets would more likely be dissimilar. The contrasting scenario, where similarity between age-matched newborn networks (i.e., DBI and dHCP) was less compared to that of pre- and full-term scans (i.e., dHCP), was observed when linear models were used (Revised Figure 7.D). Note that dHCP scans belonged to unique subjects, thus similarity will not be due to scans belonging to the same individual scanned at different times. The results suggest that the linear models may be more attuned to non-neuronal features of the scan such as recording parameters, preprocessing, and data length. Related to this, the following changes were made to the manuscript [page 17 line 364-378]:

“The opposite was observed with linear models. Specifically, the age-matched inter-dataset pattern similarity (black line in Figure 7.C) was higher than intra-dataset similarity (red line in Figure 7.C; full-term vs. pre-term in dHCP) with VAE-derived latent variables. Linear models, on the other hand, showed higher intra-dataset similarity (Figure 7.D). This observation remained consistent using varying number of top sorted ICs. The finding that the VAE revealed similar network patterns in two independent, age-matched neonatal datasets suggests that the VAE may be a better tool than linear models for capturing neurophysiologically relevant brain activity. In contrast, it is likely that ICA and cortical parcel techniques were homing in on features such as equivalent acquisition parameters, preprocessing, etc., that were similar between the pre- and full-term dHCP scans.”

Related to the developmental significance of networks revealed by the VAE model, we believe the superior age prediction performance of the VAE compared to linear models across different datasets suggests that the VAE may be capturing developmentally meaningful signals.

Regarding the comment “… the current approach did not include subcortical structures that are known to play a key role in the development of functional networks (e.g., transient developmental layers like the subplate, the thalami), which requires further analysis or discussion.”

A technical limitation of the current version of the VAE model is that it could not utilize information from the subcortical regions. This is partly due to the unique architecture of the VAE where the model receives separate brain patterns from left and right hemispheres and mirrors features from the two images during convolution, based on our knowledge of hemispheric symmetry (i.e., the primary visual cortex in the left hemisphere is in a similar location on the right hemisphere). If we wanted to include subcortical regions to the VAE model, we would need reliable estimates of (1) functional/structural aspect of L-R subcortical symmetry and (2) correspondence/connections between subcortical and cortical regions. Without these, it would be very difficult to add information from subcortical regions to the current model without rebuilding the whole model architecture. Thus, despite its role in brain development, we are unable to incorporate data from the subcortex into the model. Admittedly, this is a limitation of the current study that requires further investigation.

We now discuss this in the limitations of our study [page 25 line 558-570]:

“Lastly, one limitation is that our VAE model currently does not capture information from the subcortex which plays a critical role in neural circuitry formation^1, 2^. Currently, we are unable to add the subcortical regions into the VAE model; doing so would require rebuilding the entire model architecture to include information such as subcortical projection into the neocortex. While there are validated adult deep learning models for how cortical hemispheric information are combined during convolution (such as hemispheric symmetry in the current implementation), to the best of our knowledge, none exist for how to match subcortex data to cortex data. One possible way to circumvent this is to use the 3D volumetric rsfMRI activity as input to the deep learning models; however, as mentioned above, this could lead to significantly degraded spatial resolution. Alternatively, graph-based deep learning techniques could be explored. Other deep learning applications that could more effectively model information from subcortical regions should be further investigated.”

2. The potential of the whole approach to describe the emergence and development of functional connectivity needs to be tempered as the use of a VAE model trained on adult data is based on the strong assumption that connectome features are rather similar/stable across ages. Therefore, it cannot reveal structures/networks that would be strongly different (e.g. present at early stages but not in the adult brain). The smoothing effects inherent in VAE also make the method relatively insensitive to the fine granularity of developing networks. In addition, volume registration may be less effective than surface alignment given the intense growth and increased folding over this age range.

Regarding the assumption of shared brain networks across the lifespan:

The VAE model could only represent networks that it has been trained on, thus, by its very nature, the VAE model will fail to capture networks present in newborns that are completely absent in the adult brain. This is a limitation of the technique, and with this in mind, we revised our discussion (quoted below) to reflect a more nuanced perspective of how the VAE model could help characterize emerging brain networks.

Related to this, we would argue that this limitation (of somehow being constrained by adult data) is one shared by most, if not all, analytical models currently used in fetal-neonatal neuroimaging studies. Even though not all models are trained on adult rs-fcMRI data, most fetal-neonatal neuroimaging studies are initially approached and interpreted through the lens of human adult neuroscience. This is not ideal but is usually the case^3, 4^. One example is when newborn networks defined using ICA are selected/interpreted by matching them to known network patterns (e.g., default mode, sensorimotor, visual, auditory, etc.) in human adults. The point of influence just moves from model generation (VAE) to interpretation (ICA). We argue that utilizing adults rs-fcMRI at the level of the analytical model is more desirable than at the interpretation level, as there may be information from rs-fcMRI in adults that can more objectively guide the way we understand representations of neonatal rs-fcMRI (as shown by our findings).

The VAE can be an important tool for investigating the emergence/development of brain functional networks over the lifespan. Because it is a non-linear model, it can reveal network patterns not captured by methods such as ICA. To give an example, a primitive DMN has been shown to exist in fetuses^5^; its pattern continues to evolve with development involving more areas of the brain as the individual matures. Thus, the spatial evolution of the DMN over aging is non-linear by nature, which cannot be properly estimated by linear analytical methods. The VAE, unlike linear models, has the potential to map non-linear trajectories of fetal-neonatal neurodevelopment, by defining two distinct latent variables relevant to either functional brain networks or aging, in the non-linear latent space. Although our current study was based on the hypothesis that fetuses/neonates and human adults largely share brain activity patterns, dissimilarity over different age populations can be addressed by training/finetuning the VAE model on datasets belonging to different age groups, as currently being investigated in a follow-up study. Please see our revised discussion below [page 27 line 591-605].

“A promising application of the VAE model may be for characterizing the developing brain’s functional connectome. As the human brain develops rapidly over the fetal-neonatal periods, it is likely that the spatial pattern of FBNs will also evolve. For example, it has been suggested that a nascent default mode network (DMN) exists in fetuses ^5^, while the DMN of neonates has been shown to closely resemble the human adult ^6^. However, due to the linear nature of conventional models such as ICA, it is nearly impossible to capture the spatial evolution of FBNs over aging. For example, linear ICA can show changes in network strength but not changes in network spatial patterns with aging (see Figure 4 in ^7^). In contrast, the VAE model is non-linear such that combining two latent variables in the latent space can yield continuous change of network patterns in the cortical space (i.e., adding/subtracting age-related latent variable from DMN-related latent variable and projecting into the cortical space using the VAE decoder). Thus, the VAE model may provide important insights into fetal-neonatal neurodevelopmental growth trajectories that are inaccessible using conventional linear models.”

Regarding the “… smoothing effect inherent in VAE.…” that “also make the method relatively insensitive to the fine granularity of developing networks.”

Unfortunately, some level of smoothing is somewhat unavoidable for most, if not all, representational methods (e.g., VAE, ICA, or PCA). To illustrate this point, we added Figure 3 in the revised manuscript showing the spatial similarity between reconstructed and manually smoothed activity patterns for VAE and linear models as a function of smoothing at increasing FWHM levels. As suggested by Reviewer 3, we also compared the reconstruction performance of neonatal rs-fcMRI with young adults. The manuscript has been revised as follows:

Methods [page 33 line 742-745]:

“To evaluate the smoothing effect of VAE and linear models, we manually smoothed the original rsfMRI data using HCP workbench code -cifti-smoothing. The smoothing size was defined as full-half-maximum-width (FWHM) and varied from 1 to 10mm at 1mm intervals.”

Results [page 7 line 151-170]:

“To test whether improved representations in the VAE stemmed from a stronger model smoothing effect, we evaluated the smoothing effects of the VAE and linear models as a function of varying FWHM level (Figure 3). Briefly, we measured spatial similarity between reconstructed and manual activity patterns at different FWHM levels. The smoothing effect of the models was approximated by its peak spatial similarity (e.g., smoothing effect of the VAE = 5mm FWHM). If better reconstruction performance of the VAE model merely originated from its stronger smoothing effect, the trends of spatial similarity of the VAE and linear models over increasing FWHM levels would intersect or cross over. However, this was not observed, suggesting that the improved VAE reconstruction performance of the VAE stemmed from its superior representations of neonatal rs-fcMRI.

As expected, reconstruction performance of adult rsfMRI (blue line) was better than neonatal rsfMRI data. Also, the maximum FWHM level was at 7mm for adults compared to 5mm for neonates. Given that brain size of newborns is about 60% of adults’ brain size, this result suggested that smoothing effect of the VAE is largely consistent against age-related characteristics of rsfMRI. Similar smoothing effect was observed for IC300 and melodic IC with 256 components. IC50 showed the strongest smoothing effect as the peak was beyond our FWHM search boundary. Nonetheless, the VAE showed the best reconstruction performance compared to other linear counterparts across all smoothing levels.”

Regarding concerns about the effectiveness of volume registration:

We agree with the reviewer that projection to the cortical surface may be more desirable, but several issues hindered us from performing such analyses. We unsuccessfully searched for a reliable toolbox for reconstructing cortical surfaces of the fetal brain. Likewise, we tried FreeSurfer, but their models were currently not optimized for fetuses. Our lab is now in the process of developing such a tool, but this project is at an early stage. Thus, we performed the brain registration in the volume space. We carefully checked the quality of volume registration and confirmed that registration was reasonable for all scans except 7 scans Those scans were excluded in the analysis (regarding it, we kindly refer the reviewer to our response to R2-M1-1) In line with our DBI dataset, dHCP dataset also employed volumetric registration. We do not know the specific rationale for volumetric registration in the dHCP dataset, but we speculate their reasons may be like ours. Please see changes made in the revised manuscript [page 31 line 695-696]:

“We carefully visually examined the quality of volumetric registration and confirmed that registration results were reasonable for all scans.”

3. The performance of the reconstruction might be driven by several factors other than functional information and maturation, most of which are also age-dependent: brain size and folding (in relation to registration issue), head motion (especially within the womb), acquisition settings (in fetuses vs. neonates), scan duration, processing pipeline, etc. In addition, differences in preprocessing between the two datasets (dHCP vs. DBI) could explain part of the differences in reconstruction, prediction, and mapping findings. It is important to disentangle the effects of potential confounding factors from the developmental mechanisms to consolidate the current findings and interpretation.

We appreciate the constructive feedback. In the revised paper, we supplemented the original analyses investigating associations between possible sources of variation – brain size, head motion, scan duration, and radiology score – and reconstruction performance in the DBI and dHCP datasets.

First, we found that brain size was negatively correlated with reconstruction performance in the fetal group but not in the neonatal group in the DBI dataset (Figure 4—figure supplementary 2 D and F). After including brain size into the regression model, we found that the negative association between age and reconstruction performance was, as expected, attenuated (i.e., since age and brain size are highly correlated) (Figure 4—figure supplementary 2 E; from r=-0.69 to -0.24).

In the revised manuscript, we further evaluated the contribution of head motion to reconstruction performance in the dHCP dataset. In the original version, we reported that the head motion was negatively correlated with reconstruction performance, or that higher motion led to poorer reconstruction. Our follow-up analysis showed that the age-dependence of reconstruction performance was further increased when the head motion was considered in the regression model. Note that there was no significant correlation between age and head motion; r=0.01, p=0.79. These results were added in the revised manuscript [page 10 line 217-222]:

“In the dHCP dataset, reconstruction performance was negatively correlated with average frame-wise displacement level (r=-0.34, p<10^-4^) but not with brain size (r=-0.07, p=0.17). There was no significant correlation between PMA at scan and head motion (r=0.01, p=0.79). When head motion level was considered in the model, the age-dependency of the reconstruction performance increased (r=0.22; p<10^-4^).”

It is noteworthy that data length did not significantly contribute to variability in reconstruction performance in the dHCP dataset. This is reflected in the revised manuscript as [page 11 line 222-223]:

“The reconstruction performance remained consistent across different scan duration (Figure 4—figure supplementary 3). “

In the dHCP dataset, we also found that the reconstruction performance of neonates with a radiology score=5 (incidental finding with possible / likely significance in their MR images) was lower than groups with a radiology score = 1/2 (normal or no clinically incidental findings in MR images) (Figure 4—figure supplementary 1 in the revised manuscript). The existence of clinical MRI findings suggests greater distance from normative, baseline brain activity pattern and was likely a significant factor in the poorer reconstruction performance. This supports our original argument that the lower (or higher) reconstruction performance could be a surrogate measure of the resemblance of fetal-neonatal activity pattern to human adults. Below are changes made in the revised manuscript [page 9 line 183-191]:

“We also observed that subjects with radiological scores=5 (incidental finding with possible / likely significance for both clinical and imaging analysis; n=23; mean ± SD=0.42 ± 0.05; two-sample t-test, *p*=0.046 vs. group with score=1; n=172; mean ± SD=0.44 ± 0.05) showed worse reconstruction performance compared to other groups (Figure 4—figure supplementary 1). When only scans with good radiology scores (=1 or 1+2) were considered, the association between age at scan and reconstruction performance remained significant (group with radiology score=1, normal appearance for age; n=172, *r*=0.19, *p*=0.01; score=1+2, 2=Incidental findings with unlikely significance for clinical; outcome or analysis; n=292, *r*=0.21, *p*<10^-3^).”

In the discussion [page 21 line 451-453]:

As the VAE was trained to represent resting-state brain activity of neurotypical adults, lower (or higher) reconstruction performance in fetuses or neonates can be interpreted as lower (or higher) similarity of their FC representations to the adult brain. This is supported by our finding of lower reconstruction performance in neonates with incidental MRI findings with possible / likely clinical significance (radiology score = 5) compared to other groups.

The above findings along with the findings reported in the original manuscript collectively suggest that factors including brain size, head motion, and radiology score (but not scan duration) were significantly associated with reconstruction performance; after accounting for these factors, however, the relationship between age and reconstruction performance remained significant. In the original version of the manuscript, we also assessed the influence of brain size and head motion on the inter-scan similarity of reconstruction performance found in the dHCP.

However, despite our best efforts to identify factors relevant to variations in reconstruction performance, many other factors remain untested due to technical issues or limited data. We want to emphasize, however, that the primary goal of our study was to demonstrate the advantages of VAE over linear models in analyzing fetal-neonatal rs-fcMRI data in datasets acquired using different acquisition parameters. While we evaluated several factors that could account for differences in reconstruction performance between the two datasets, we believe this problem warrants further scrutiny that would be best served by a separate, more focused study. We hope the reviewers agree.

Reviewer #1 (Recommendations for the authors):There are some vague expressions and potential methodological biases throughout the manuscript, which should be further addressed prior to publication.1. The authors emphasized representing the non-linear maturation of brain functional maps from fetal to neonatal stage by their methods throughout the manuscript. But they failed to interpret the contribution of the proposed approach for capturing this non-linear development. Their model is pre-trained by adult data. The non-linear effect of variational autoencoder refers to the non-linear combination of representing fMRI signals between brain vertexes at the individual level. The age prediction is performed using the SVM model. None of these is directly related to non-linear development. Which result and analysis support this hypothesis?

We appreciate the reviewer’s critical comments. We addressed this issue in our response to the Reviewing Editor’s comments. To reiterate, the editors and reviewers make an important point and we have now modified several segments of the manuscript. Now, we highlighted VAEs advantages over linear models and discuss its potential for capturing non-linear maturation of brain functional maps during the fetal to neonatal periods. In the main text [page 27 line 591-609], we provided our perspective on how the latter could be achieved using VAE.

2. Some potential biases in the comparison with linear ICA methods in age prediction analysis should be further evaluated. First, the number of latent variables and the number of ICA components should be kept the same during the comparison. Second, the effect of individual functional mapping on the baby brain should be taken into account.

The reviewer raises an important point. To address this, we have now evaluated reconstruction performance using 256 components, matching the number of VAE latent variables. For this, we used melodic ICA. Melodic was applied to the training set of the VAE model. We found that the reconstruction and age performance of Melodic was significantly worse than VAE. The following changes were made in the manuscript:

Methods [page 38 line 846-848]:

“Among them, we utilized IC maps consisting of 50, 100, 200, and 300 ICs (named as IC50, IC100, IC200, and IC300). To match the number of latent variables, ICA maps with 256 components (named as melodic ICA) were estimated by applying melodic ICA^8^ to HCP adult data that were used for training the VAE model.”

Results [page 7 line 142-144]:

“This observation remained consistent at the group level and across different datasets (Figure 2. E). Interestingly, IC maps with higher (=300) or equal (=256; MelodicIC) dimensions than VAE (=256) showed inferior reconstruction performance than VAE model (Figure 2. E).”

MelodicIC results were also added in Figure 2.E.

We also assessed the age prediction performance of melodic IC. Like the linear models reported in the original version, Melodic performance was inferior to that of the VAE model. We included the results in Tables 1 to 3.

3. The advantages of the VAE approach in generating brain functional networks at the system level are not exhibited. Do their methods find more meaningful functional systems than previous studies? This should be well evaluated.

We thank the reviewer for the critical comment. In the revised manuscript, we have included more results that, we hope, can address the reviewer’s concern. We think one strength of the VAE model compared to linear models is that it can generalize across datasets and ages (i.e., adults vs. full-term newborns vs. preterm infants vs. fetuses). In the original manuscript, we supported this point by showing the superior age prediction performance in both DBI and dHCP data, suggesting that the VAE, more than linear models, identifies features relevant to brain development. In addition, we also included analysis showing that only the VAE model can extract latent variables representing brain networks across different datasets. We discussed this in depth in our response to the Reviewing Editors inquiry about the biological significance of networks revealed by the VAE; we invite the reviewer to read our response to **RE-M1** and view the revised Figure 7D here [p3].

Accordingly, the following changes were also made in the Discussion section [page 24 line 523-526]:

“Related, we found that FBNs defined by the linear models (IC50, IC200, and IC300) were more sensitive to inter-dataset difference, resulting in lesser pattern similarity between dHCP and DBI full-term groups, and greater similarity between pre- and full-term newborns within the dHCP dataset (Figure 7D).”

Reviewer #2 (Recommendations for the authors):The paper is a very interesting application of the VAE method to the early brain. This represents an important amount of work that went into the analysis and presentation of the results. The work is easy to navigate and clearly written (some small problems are detailed below), although the supplementary information cited within the text was not made available. The figures are of good quality and clear to interpret.• Both data sets have a fairly large range of PMA at scan (in DBI for foetuses from 20w GA). Fast and large-scale changes are taking place within the developing brain in this period which might affect the data processing pipeline at all stages. Please include more details if any quality checks were performed. E.g.:– Any rsfMRI QC for dHCP data;

Quality checks were performed for both dHCP and DBI datasets. For the dHCP, echoing the response to Reviewer1-M16, we carefully reviewed the scan quality summary provided by the dHCP (file name: sub-SUBID_ses-SESSID_funcqc.html) and added this detail to the manuscript [page 31 line 687-690].

As a side note, we refer the quality check performed in the DBI dataset, resulting in exclusion of 7 scans. The change was updated in the main manuscript [page 30 line 655-657]:

“To that end, 21 neonatal scans (14 with short scan data <4mins and 7 with partial or full failure at volumetric registration) among 160 scans were excluded in the subsequent analysis.”

– For dHCP, babies were included independently of their radiological scores -> this might have a significant impact on the reconstructed functional data;

The reviewer’s hypothesis is indeed correct. We analyzed the role of radiology scores on reconstruction performance and found that higher scores or the presence of incidental findings on MRI was associated with lower reconstruction performance. In the interest of brevity, we invite the reviewer to peruse our response to **RE-M3** [page 12] where we presented the results and quoted revisions to the manuscript. Changes related to this analysis in the manuscript can be found here [page 9 line 183-191] and [page 21 line 451-453].

– How good was the projection to cortical space? (Authors use 40w template, there is a large temporal distance for some babies – there might be errors distributed non-randomly and based on ages);

For the DBI dataset, we carefully inspected the quality of projection to cortical space and our visual inspection revealed reasonable projection quality across all scans included in the study, including the younger fetuses. To minimize possible errors due to projection to cortical space, we employed a two-step approach: (1) the individual brain template was projected to the age-matched brain template, and (2) the output from #1 was projected to the 40-week brain template. This two-step procedure, instead of directly projecting the subject’s brain to the template, was believed to reduce the projection errors as suggested elsewhere in dHCP ^15^. We also point out here that between-scan similarity of reconstruction performance was evaluated after accounting for GA/PMA; thus, we think that it was unlikely that projection errors significantly impacted our observations. Still, we acknowledge that qualitative evaluation of images has its limits and that we did not test for effects of projection directly. Thus, we added the paragraph below to our discussion to remind readers of some of the study’s constraints [page 21 line 453-461].

“It needs to be pointed out, however, that the age-dependency of reconstruction performance could be interpreted in other ways. It is possible that errors were introduced when projecting fMRI in individual brain space into the 40-week standard brain template. In an attempt to reduce errors, we used a two-step approach for the registration: first, registering from the subject space to the age-matched brain space and second, from the age-matched brain space to the 40-week template space. We accounted for the effect of brain size in the regression model. Thus, we believe our observation was minimally biased by projection errors; this, however, needs to be confirmed.”

– Geometric reformatting – errors with inflation, 3D->2D?, if, are those distributed randomly (unlikely);

As the reviewer suspected, geometric reformatting does introduce geometrical error that is most pronounced at the top and bottom of the cortex. In our previous study with human adults (see supplementary Figure 1 in ^9^), we investigated the amount of information loss due to geometric reformatting and concluded that the amount of error was minimal compared to the error due to compression/reconstruction procedure. We measured error due to geometric reformatting in the current dataset and concluded that it is unlikely that it accounted for the association between age and reconstruction performance. In the manuscript, we detailed our analysis and perspective [page 21 line 461-466]:

“Another possibility is that error originating from geometric reformatting accounted for the age-dependency of reconstruction performance. We think that this was unlikely as, in our previous experiment with human adults (see supplementary Figure 1 in ^9^), we showed that the error induced by geometric reformatting was minimal (*r* ~0.98 between the original cortical pattern and reformatted/recovered cortical pattern).”

– During dHCP data description authors say 'The additional step of voxel-wise detrending, bandpass filtering and voxel-vise normalization' after volume-to-surface mapping. Were these steps applied to DBI set as well?

We did apply voxel-wise detrending and bandpass filtering to the DBI dataset. After preprocessing, voxel-wise normalization (setting to mean as 0 and standard deviation to 1) was applied. To clarify the DBI data preprocessing, we revised the methods as follows [page 29 line 645-648] and [page 30 line 660]:

“Functional MR images were preprocessed as follows: slice time correction, discarding the first four volumes, de-spiking, bias-field correction, motion-correction, intensity scaling, detrending, bandpass filtering (0.009-0.08 Hz), and nuisance regression with motion parameters.”

“Preprocessed rsfMRI scans at the volumetric brain space were projected to the standard cortical space using HCP workbench command *-volume-to-surface-mapping*. The timeseries for each voxel was then normalized (zero mean and unitary variance).”

– Overall, more detail on the processing pipeline(s) would be useful with details on possible differences between datasets and focus on limitations that stem from it on the interpretation of results.

We thank the reviewer for the suggestion. Supplementary File 1a showing the preprocessing differences between datasets has now been added [page 29 line 652-653].

“Preprocessing steps for the fetal and neonatal MRI data can be found here ^4^ and here ^50^, respectively. The preprocessing steps applied to the two datasets (dHCP and DBI) are summarized in Supplementary File 1a.”

As suggested by the reviewer, we discussed the possibility that the difference between two datasets stem from difference in preprocessing steps as below ([page 23 line 515-519] in main manuscript):

“In general, we observed that FBNs identified in the DBI dataset were relatively less straightforward to interpret, compared to the dHCP dataset, possibly due to many factors including, but not limited to, smaller scan size, shorter data length, different preprocessing steps such as ICA-FIX, or a combination of the above (see Figure 7—figure supplementary 1 and 2 for the whole set of FBNs from dHCP and DBI datasets).”

• It would be useful to perform additional data analysis between the two datasets before there are forwarded to the VAE – is there a significant difference between datasets that are due to the pre-processing? Could these explain differences in reconstruction, predictive, and mapping differences?

We appreciate the reviewer’s critical comments. To clarify, we included two datasets that were acquired and preprocessed differently in the analysis to demonstrate the generalizability of representations defined with the VAE model. For this reason, in the original manuscript, we did not focus on the differences between the two (although initial analysis exploring differences between the two datasets have now been included; see our response to **M1-6**). Instead, the scope of our current work lies on the reproducibility of our findings across the two datasets. We do agree with the reviewer that comparisons between the two datasets in terms of their performance on different tasks such as age prediction, reconstruction, or network mapping would be very interesting and informative. However, given the complexity of these topics, a thorough comparison would be best served by a separate study. To prevent confusion regarding the scope of the current work, we removed this statement from the original manuscript:

“Age prediction in the dHCP dataset was better than the DBI dataset; this is likely due to the shorter data duration (4-6mins in DBI datasets vs. 11 mins in dHCP dataset) in the latter.”

• It would be very useful to focus on the performance of the VAE compared to adults given the method was developed and validated in adults and most of the novelty comes from its re-application to infants. Please, detail how the reconstruction compares in DBI, dHCP vs adults. It would be good to think about other interpretations for this difference on top of the neurodevelopmental stage (approximated by PMA at scan)- brain size, projection to the template, geometric reformatting to 192x192 array.

As the reviewer suggested, we compared the reconstruction performance between DBI newborns and adults. As expected, the reconstruction performance in adults, the age group that the VAE was trained on, was superior to newborns. Our updated observations are summarized in a new figure (Figure 3 in the revised manuscript; or see our response to RE-M2 [page 7]) showing the smoothing effect of the VAE and linear models as function of varying FWHM level (x-axis).

A detailed response can be found here response to R1-m5 (brain size); R2-M1-3 (projection to the template); R2-M1-4 (geometric reformatting); we kindly request that the reviewer read our response to Reviewer 1, who shared similar concerns. Changes related to this in the revised manuscript can be found here [page 21 line 453-466].

• Correlation of reconstruction error with PMA at scan (Figure 3).– Figure 3B: It would be good to incorporate more detail on the 3 outlier points in DBI dataset – why does it fail so much – do these kids happen to have low quality recording? Or maybe injury that makes the model fail? Understanding why it fails might give informative cues on what the model is actually trying to represent.

We appreciate the reviewer’s constructive comments. It was highly helpful for us to re-evaluate the overall quality of our dataset. Once again, we carefully double-checked the quality of projection. To that end, we found that 7 scans among 139 neonatal scans in the DBI dataset showed partial failure at volumetric registration. We further confirmed that 3 outliers in the original studies were included in 7 subjects identified here. Therefore, we excluded 7 scans from the whole analysis, and we found there were no outliers in the revised results. A detailed update can be found in our previous response to M1-1 or [page 30 line 661-663] in the revised manuscript.

– Also, the 3 outlier points in DBI dataset are in the 'normal' cloud in dHCP dataset, please discuss why the reconstruction might be so different between DBI and dHCP datasets

Please note that the outliers in the original figure have been excluded in the revised paper (as discussed in the previous question). As mentioned previously, a detailed comparison between the two datasets was not the primary goal of the current work. Still, we do want to share our perspective on the reconstruction performance differences between the two datasets given the importance of this issue. We thought of several candidate factors including data length, preprocessing steps, and recording parameters. During the revision process, we found that the reconstruction performance was not driven by differences in scan duration (Figure 1—figure supplementary 3; also see response to RE-M3 [page 10] and updates in the manuscript here [page 11 line 222-223]). This left us with, at least, two factors: (1) different spatial resolution, 2.15mm isometric for dHCP vs. 3.125×3.125×3mm for DBI, and (2) different preprocessing steps, for example, ICA-FIX. We intend to investigate these factors in our future work.

– Please show reconstruction with age for the foetal data. Discuss more why this negative correlation – it can be smoothing, differences in processing, but also movement etc. – in any case, these points make us expect the relationship to be weaker but not to be strong and in the reverse direction, comment more on the foetal processing pipeline

We appreciate the opportunity to clarify. We believe that the smaller brain size in younger fetuses significantly contributed to the negative correlation between age and reconstruction performance in the fetal data. We kindly ask the reviewer to read our detailed response to reviewer R1-m5 here, and view changes in the revised manuscript here [page 9 line 191-213]. We summarized analyses related to this topic in Figure 4—figure supplementary 2.

Regarding the validity of fetal preprocessing pipeline, our previous studies^12, 16^ have investigated fetal neurodevelopmental characteristics and our findings were in line with other studies that employed independent datasets. Given that, we speculate the choice of our preprocessing step for the fetal rsfMRI was not the main factor for our findings. We hope the reviewer is amenable to our perspective.

– You are taking head motion into account between sessions – however there might be a relationship between motion and age => it would be good to re-run the same analysis as in reconstruction vs age but with motion

We reported the results for this analysis in the original manuscript. There was no significant correlation in the inter-session motion. After accounting for head motion, the inter-session similarity of reconstruction performance remained significant. Please see the discussion here [page 11 line 234-242] in the main manuscript; Figure 4—figure supplementary 4 summarize these findings.

– It would be interesting to see the same comparison for the ICA – e.g. what are the correlations for the second best method IC300? Is that capturing more age/individual signature/noise? If the difference is significant would be a supporting argument for improvement using VAE.

As suggested by the review, we compared the inter-dataset similarity of network patterns derived using ICA. Unlike the VAE, network patterns derived by ICA showed the best similarity within the same dataset (preterm vs. full-term in the dHCP). To demonstrate this, we quantified the difference between inter-dataset pattern similarity (between DBI neonate and dHCP neonate) and intra-dataset pattern similarity (between dHCP neonate and dHCP preemie). We observed that only the VAE showed higher inter-dataset pattern similarity versus intra-dataset pattern similarity while linear counterparts (IC50, IC200, and IC300) showed higher intra-dataset pattern similarity. As the reviewer pointed out, the above results suggest that VAE-derived latent variables, unlike linear models, captured network-related features that were generalizable across different datasets. Collectively, our results suggest that the VAE model is better suited than linear models for multi-dataset, large-scale analysis. For changes in the main manuscript, we invite the reviewer to read our response to RE-M1 [page 3].

• In the age prediction methods, please indicate final feature set sizes after feature selection to give an idea of what proportion of the latent space might be representing age information, what is the interpretation for the features not associated with age? (if individual signature, could you look into the predictive power of the latent vector to identify the individual in the future?)

As suggested by the reviewer, we examined the percentage of latent variables significant for age prediction in the training datasets across different latent spaces. Briefly, we found that VAE showed the most consistent percentage of latent spaces involved to age prediction, between DBI and dHCP datasets, compared to other linear models. We speculate that this consistency, at partially, plays in the role for the superior inter-dataset age prediction accuracy of the VAE. Please see changes made in the revised manuscript:

For dHCP [page 12 line 262-265]:

“The percentage (=14.14±0.32%) of latent space representing age information was lower than most of linear counterparts (for cortical space, 76.53±1.29%; for IC50, 44.84±0.46%; for IC100, 32.55±0.43%; for IC200, 16.07±0.34%; for melodicIC, 29.97±0.47%), except IC300 (=10.83±0.25%). “

For DBI [page 13 line 285-288]:

“The percentage of VAE-derived latent space representing age information (=11.39±0.35%) was lower than IC100 (=10.86±0.27%), IC200 (=7.31±0.25%), and IC300 (=5.24±0.23%); comparable to cortical parcels (=11.12±0.24%); higher than IC50 (=21.68±0.43%) and melodicIC (=12.6±0.24%).”

Finally, the updated results were discussed on [page 19 line 415-420]:

“Like morphological distortions in natural images, fMRI images, despite efforts to spatially and temporally denoise the data, may still carry non-neural signals. Supporting this notion, we found that the percentage of VAE-derived latent variables sensitive to age prediction in dHCP dataset was comparable to the one observed in the DBI dataset (14.14±0.32 vs. 11.39±0.35%, for dHCP and DBI), whereas larger inter-dataset discrepancy were observed with linear models (most difference was in cortical parcel; 76.53±1.29 vs. 11.39±0.35% and least difference was in the IC300; 10.83±0.25 vs. 5.24±0.23%).”

Regarding the interpretation of features not associated with age, we are currently working on identifying FC profiles relevant to age prediction versus individual uniqueness. We invite the reviewer to read our responses to the RE-M1 found here [page 3] and R1-m13, where we discuss our approach and report our initial findings. As a short summary, consistent with the reviewer’s idea, we found that age-insensitive latent variables identified individuals in a subject identification task better compared to age-sensitive latent variables.

• Regarding comments on summing the significantly correlated features (again specify how many) 'to make model more robust against different recording specifications' and then L203: 'age prediction in the dHCP dataset was better than in the DBI dataset, this is likely due to the shorter scan duration' => this makes if for an unfair comparison, in the summing – the latent space is collapsed into one feature which will necessarily remove information, additionally, the reconstruction of the DBI data (specifically foetuses) showed strange relationship with age. Moreover, the processing was different (with its own difficulties for foetuses). There are more limitations other than shorter scan durations involved, please discuss these

Again, we thank the reviewer for the constructive comments. We agree with the reviewer that there are numerous factors beyond scan duration that could be impacting age prediction in the DBI and dHCP datasets. In line with this, we have now removed the statement “age prediction in the dHCP dataset was better than in the DBI dataset” from the manuscript. Other factors that may play a role in age prediction performance differences, are different recording parameters, different preprocessing steps, and different age prediction algorithms, to name a few.

For comments on summing significantly correlated features, please see our response below and changes in the manuscript here [page 35 line 772-778].

For comments on the negatively correlated relationship between GA and reconstruction performance in fetuses, please see the response to R1-m5 and changes in the manuscript here [page 9 line 191-213].

Regarding the preprocessing step of fetal rsfMRI, we kindly refer the reviewer to the response to R2-M1-2 and R2-M4.

• Please detail how the prediction fares on neonates vs foetuses in light of reconstruction with age relationships being very different

The reviewer brings up an important point. Considering the purpose of our analysis was to prove the superiority of the VAE over linear counterparts (as opposed to comparing VAE performance on the two datasets), we believe that it is reasonable to perform the age prediction task in the two age groups if the identical analytical framework was applied to the different latent variables. Nonetheless, we also performed the age prediction in separate age group, as suggested in the next comment. Please refer to the response below regarding the age prediction performance for each age group. In addition, note that our choice of merging the feature set of age prediction model in the DBI dataset was to minimize possible bias due to factors such as different recording parameters/different preprocessing steps/age-associated pattern. To further clarify, we revised the justification for the age prediction model used in DBI dataset. In the Method section [page 35 line 772-778]:

“The prediction model for the DBI dataset was nearly identical to the above prediction model in dHCP dataset, except that, in the feature selection step, we took the feature positively correlated to the age and summed the significant features. We modified this step to make the model more robust against possible bias due to different recording parameters (e.g., spatial resolution) in the fetal and neonatal groups. Moreover, during reconstruction, we observed that the association between age and reconstruction performance trended in opposite directions in the neonates and fetuses. Thus, to ensure that the age prediction model was applicable in both datasets, we modified the feature selection step. As the dimension of the feature set became 1, we employed a linear regression model using MATLAB function fitlm.m. Note that, except for the feature selection step and regression model type used, all other steps were identical to those used in the dHCP dataset.”

• L278 makes it sound the evaluation was made at the age group level separately, this is not the case but would be useful. Overall, it would be beneficial to first restrict the analysis to a narrow age range, for example, the term-equivalent age with the homogeneous data acquisition settings and processing steps to validate the application to dissociate whether the results really do capture the variation within the functional connectome rather than all the other potential confounders.

Based on the reviewer’s suggestion, we performed age prediction performance within the same age group (neonate and fetus) in the DBI dataset. Regarding the age prediction model, we revised the methods section as follows [page 35 line 781-787]:

“We also performed age prediction on neonatal and fetal datasets of the DBI separately. For this, we modified the prediction model in two ways. Since the number of samples was limited compared to the combined fetus/neonate model, we skipped the step adjusting for age prediction bias; thus, 90/10% of data was used for the training/testing set, respectively. Another modification was adjusting the feature selection threshold from FDR-corrected *q*<0.05 to uncorrected *p*<0.01.”

As shown in Supplementary File 1b, we found that VAE model outperformed other linear counterparts in the neonate age group, but cortical parcel showed the best performance in the fetal group. This revised result suggests that the superior age prediction performance of VAE model becomes more pronounced when examining a wider range of age.

Accordingly, the result was updated to [page 13 line 282-285]:

“When the age prediction model considered a single age group (neonates or fetuses), the age prediction accuracy of VAE was superior to linear models only in fetuses; cortical parcel showed better performance in fetuses (Supplementary File 1b).”

Finally, in the Discussion, the following changes were made [page 19 line 420-427]:

“In fact, when relatively narrower age variation was considered in the age prediction task (only fetus or neonate) in the DBI dataset, the VAE showed the best age prediction performance only in neonate; cortical parcel showed the best age prediction accuracy of fetal group. Nevertheless, the cortical parcel completely failed to predict the age across different datasets (MAE = 12.06±15.35 weeks) whereas VAE showed the best generalizability of age prediction across different datasets, MAE = 3.90±0.65 weeks (Figure 6 and Table. 3).”

• L250: does it make sense the author found precursor of the default mode network in DBI and not in dHCP given the DBI subjects are generally younger than dHCP and thus further from the cited young adults rsfMRI studies?

Upon careful review, we also observed the default mode network in the dHCP dataset. Therefore, we updated our results and interpretations. We also found that DBI dataset showed an interesting activation pattern involving the auditory, visual, and sensorimotor network but with de-activation in the temporoparietal junction region. FBN patterns of neonates are shown in Figure 7—figure supplementary 1 and 2, please see here [page 15 line 331-332]. We included these FBN findings in the results [page 15 line 332-350]:

“For example, IC8 in both the dHCP and DBI datasets showed activation/de-activation patterns between left and right early visual area. This contrast (i.e., activation vs. de-activation) was also observed at the networks level; auditory vs. somatosensory (IC6 in dHCP; IC3 in DBI) and visual vs. auditory (IC3 in dHCP; IC4 in DBI). We also identified the neonatal auditory network having bilateral activation in IC3, for both the dHCP and DBI dataset. Lastly, we observed precursors of the default mode network (dHCP; IC29, DBI; IC21; Figure 7. B right bottom), which is coincident with findings in young adult rsfMRI studies^19, 34, 35^. At the same time, there were brain network patterns that were observed only in the dHCP dataset. For example, we found inter-hemispheric opposition of the auditory network (IC22, Figure 7—figure supplementary 1) and IC14 showed bilateral activation covering the higher visual network (Figure 7. B left bottom). Global synchrony patterns, which are tightly related to global signal in fMRI data, was observed in the dHCP dataset (IC5, Figure 7—figure supplementary 1), but not in the DBI dataset (Figure 7—figure supplementary 2). In the DBI dataset, we found simultaneous activation of somatosensory, auditory, and visual networks with deactivation of temporoparietal junction that integrates sensory inputs (IC6, Figure 7—figure supplementary 2). FBNs of fetuses and preterm babies were shown in Figure 7—figure supplementary 3 and 4, respectively. Lastly, we found that our observation was consistent regardless of threshold level of cortical maps, by exemplifying one FBN (IC3 from the dHCP dataset) across varying threshold levels (Figure 7—figure supplementary 5).”

• L254-256: shorter data length is only one of the potential interpretations, another alternative (among others) might be that the maps are not reliable and mostly representing the noise in the data. In the discussion it would be good to go deeper into differences between the networks between the datasets and age groups and whether these make sense in light of the expectations. Can we expect to find the same FBNs across the lifespan? Are the ones identified early on less complex/ordered, or involving primary networks rather than associative ones?

We appreciate the reviewer’s comment. We kindly refer the reviewer to our response to RE-M3 [page 10] regarding the changes made in the discussion related to this concern.

However, we respectfully disagree that FBNs found in the DBI largely represent the noise. During the revision process, we observed that newborn FBNs found in dHCP were reproduced in the age-matched DBI dataset. As two datasets were recorded under different recording parameters and underwent different preprocessing steps, it is highly unlike two datasets share similar noise characteristics. Regarding the detailed results and our perspective, please see our response to RE-M1 [page 3] or [page 17 line 364-378] in the main manuscript.

Regarding the same FBNs across the lifespan, we believe the FBN patterns will likely evolve dramatically across the lifespan, especially during the fetal-neonatal stage. Regarding this perspective, please see our response to RE-M2 [page 6].

• L263: 'least pattern similarity was observed between preterm neonates and fetuses in the DBI dataset'. This is interesting – is there difference between PMA at scan between these two groups? In light of this observation, does it make sense to include preterm subjects in the age prediction models – and other interpretations (rather than keeping them separate)? Similarly, does it make sense to include fetuses given differences in acquisition settings and post-processing? This brings back the earlier suggestions to initially keep the whole analysis to a narrow range of heterogeneous data to validate the method && increase the confidence that the extracted maps are 'real' rather than artefact of processing.

The age range of each group has now been added to the revised manuscript as [page 15 line 320-323]

“The age ranges of full-term in the dHCP and DBI datasets were 41.05±1.76 and 41.71±1.78, respectively. The ages of preterm in the dHCP dataset and fetus in the DBI dataset were 34.36±1.85 and 33.65±4.01, respectively.”

As suggested, we also conducted age prediction within neonatal or fetal group. Unlike fetal-neonatal group, in the fetal group, we found that the age prediction accuracy of VAE was comparable to linear models (cortical parcel showed the best performance). Note that the VAE showed the best performance in the neonatal group. We refer to R2-M8 [p50] regarding the updated results.

References

1. Fransson P, et al. Spontaneous brain activity in the newborn brain during natural sleep—an fMRI study in infants born at full term. Pediatric research 66, 301-305 (2009).

2. Lordier L, et al. Music processing in preterm and full-term newborns: a psychophysiological interaction (PPI) approach in neonatal fMRI. Neuroimage 185, 857-864 (2019).

3. Rajasilta O, et al. Resting‐state networks of the neonate brain identified using independent component analysis. Developmental Neurobiology 80, 111-125 (2020).

4. Asis-Cruz D, Barnett SD, Kim J-H, Limperopoulos C. Functional Connectivity-Derived Optimal Gestational-Age Cut Points for Fetal Brain Network Maturity. Brain sciences 11, 921 (2021).

5. Thomason ME, et al. Age-related increases in long-range connectivity in fetal functional neural connectivity networks in utero. Developmental cognitive neuroscience 11, 96-104 (2015).

6. Doria V, et al. Emergence of resting state networks in the preterm human brain. Proceedings of the National Academy of Sciences 107, 20015-20020 (2010).

7. Eyre M, et al. The Developing Human Connectome Project: typical and disrupted perinatal functional connectivity. Brain 144, 2199-2213 (2021).

8. Beckmann CF, Smith SM. Probabilistic independent component analysis for functional magnetic resonance imaging. IEEE transactions on medical imaging 23, 137-152 (2004).

9. Kim J-H, Zhang Y, Han K, Wen Z, Choi M, Liu Z. Representation learning of resting state fMRI with variational autoencoder. NeuroImage 241, 118423 (2021).

10. Beckmann CF, Mackay CE, Filippini N, Smith SM. Group comparison of resting-state FMRI data using multi-subject ICA and dual regression. Neuroimage 47, S148 (2009).

11. Goodfellow I, et al. Generative adversarial networks. Communications of the ACM 63, 139-144 (2020).

12. De Asis-Cruz J, et al. Global Network Organization of the Fetal Functional Connectome. Cerebral Cortex 31, 3034-3046 (2021).

13. Gilmore JH, Knickmeyer RC, Gao W. Imaging structural and functional brain development in early childhood. Nature Reviews Neuroscience 19, 123-137 (2018).

14. Cao M, Huang H, He Y. Developmental connectomics from infancy through early childhood. Trends in neurosciences 40, 494-506 (2017).

15. Fitzgibbon SP, et al. The developing Human Connectome Project (dHCP) automated resting-state functional processing framework for newborn infants. NeuroImage 223, 117303 (2020).

16. Kim J-H, De Asis-Cruz J, Cook KM, Limperopoulos C. Gestational age-related changes in the fetal functional connectome: in utero evidence for the global signal. Cereb Cortex, (2022).

17. Fair DA, et al. A method for using blocked and event-related fMRI data to study “resting state” functional connectivity. Neuroimage 35, 396-405 (2007).

18. Anticevic A, et al. Comparing surface-based and volume-based analyses of functional neuroimaging data in patients with schizophrenia. Neuroimage 41, 835-848 (2008).

19. Klein A, et al. Evaluation of volume-based and surface-based brain image registration methods. Neuroimage 51, 214-220 (2010).

20. Toosy AT, Ciccarelli O, Parker GJ, Wheeler-Kingshott CA, Miller DH, Thompson AJ. Characterizing function–structure relationships in the human visual system with functional MRI and diffusion tensor imaging. Neuroimage 21, 1452-1463 (2004).

21. Eichele T, et al. Prediction of human errors by maladaptive changes in event-related brain networks. Proceedings of the National Academy of Sciences 105, 6173-6178 (2008).

22. Wang L, Hermens DF, Hickie IB, Lagopoulos J. A systematic review of resting-state functional-MRI studies in major depression. Journal of affective disorders 142, 6-12 (2012).

23. Evans KC, Wright CI, Wedig MM, Gold AL, Pollack MH, Rauch SL. A functional MRI study of amygdala responses to angry schematic faces in social anxiety disorder. Depression and anxiety 25, 496-505 (2008).

24. Liang M, et al. Widespread functional disconnectivity in schizophrenia with resting-state functional magnetic resonance imaging. Neuroreport 17, 209-213 (2006).

25. Finn ES, et al. Functional connectome fingerprinting: identifying individuals using patterns of brain connectivity. Nature neuroscience 18, 1664-1671 (2015).

26. Gray JR, Chabris CF, Braver TS. Neural mechanisms of general fluid intelligence. Nature neuroscience 6, 316-322 (2003).

27. De Asis-Cruz J, et al. Functional brain connectivity in ex utero premature infants compared to in utero fetuses. NeuroImage 219, 117043 (2020).

28. Bouyssi-Kobar M, et al. Third trimester brain growth in preterm infants compared with in utero healthy fetuses. Pediatrics 138, (2016).

[Editors' note: further revisions were suggested prior to acceptance, as described below.]

The manuscript has been improved but there are some remaining issues that need to be addressed, as outlined below:We all acknowledge that your revisions have significantly improved the methodological scope of the study, but the methodology described is not new and the neurodevelopmental insights still seem insufficient given the scope of eLife. The potential for a future study is mentioned, but it is such beginnings of evidence that we would like to read in this manuscript. This being said, we would like to offer you a new revision if you believe you'll be able to put more emphasis on some neurodevelopmental findings. Here is a suggestion of analysis that would seem relevant to us.As we expect that distinct functional networks might evolve differently with development, you might consider different groups of neonates according to age (not fetuses for which results are less clear), highlight between-groups differences in the estimated weights of latent features derived within the autoencoder, and provide a characterization of the latent features showing the highest contribution differences in relation to age (it would be helpful to somehow relate these features to functional networks that can be visualized in terms of brain spatial topography). In our opinion, such analysis would provide some insights on the functional features and networks that show dramatic changes during this developmental period, although such features are derived from the adult brain.

We appreciate the constructive feedback. We performed the suggested analyses and revised the manuscript to now highlight temporal changes in VAE-derived functional connectivity profiles. We think the described neurodevelopmental differences in young vs old newborns enhances the manuscript’s fit with the *eLife* journal. We hope the revisions meet the reviewers’ expectations.

We added a new result showing cortical network differences between young and old neonates (revised Figure 8 and Figure 8-supplementary figure 1). Briefly, we investigated whether the variance across 30 functional networks – measured from the time series of each independent component – defined in the dHCP dataset correlated with postmenstrual age (PMA) at scan. From the entire newborn cohort, we created subsets of young and old neonates based on their PMA at scan: young (n=41, <39 weeks) and old (n=56, >43 weeks). The remaining subjects between 39 and 41 weeks (n=287) were considered as the median group. The functional network patterns for all age groups were examined and presented in the revised manuscript, and in-depth comparisons were performed between the young and old groups. The rationale behind splitting the cohort into three, as opposed to splitting at the median to yield two groups, was to better resolve age-dependent changes in functional network patterns given the age distribution of the dHCP sample. We also demonstrated between-group differences in the spatial topology of these functional networks using dual regression. All three groups’ network patterns were similar to the entire newborn cohort but with some notable regional difference in young vs old neonates. Cortical network differences between young and old groups were computed from both cortical- and latent bases. Please see below for additional details in the sections updated in the manuscript [page 17 line 365-400]:

“2.6. Age-related group differences in VAE-defined networks

Lastly, to begin to understand how functional networks evolve as the brain matures during the newborn period, we investigated differences in non-linear latent variables between young and old neonates. […] Altogether, our results highlight that the VAE can identify spatial aspect of neurodevelopment during perinatal period, which are difficult to be achieved by conventional linear counterparts..”

If you are not convinced by this suggestion, please propose another analysis that you think would be more relevant to show some insights on the developing functional connectivity from your methodology.

We agree with the reviewers’ and performed the suggested analyses (please see above). Thank you for the thoughtful feedback.

It would also be very welcome to emphasize that a future study should provide a new training procedure in babies and integrate sub-cortical structures, in order to highlight features potentially present in the developing brain but not in the adult brain.

Thank you for the suggestion. Based on your feedback, we added to our previous discussion on the lack of subcortical data in the current model [page 27 Line 605-609]:

“Another limitation is that our VAE model currently does not capture information from the subcortex. This region plays a critical role in neural circuitry formation and including subcortical neural activity in future studies would greatly enhance our understanding of the developing functional connectome^3, 4^.”

We also added statements on the importance of incorporating newborn rsfMRI data in future models [page 28 Line 620-622]:

“Future studies incorporating newborn rsfMRI data would likely help emphasize features uniquely present in the developing brain.”

Finally, we would like to stress that multiple rounds of revisions are not usual in eLife, thus we hope that you will be able to provide a new manuscript fitting with our expectations. Otherwise, we would advise you to submit your work to another journal with more focus on methods.Reviewer #2 (Recommendations for the authors):We would like to commend authors for the amount of additional work performed in response to the initial round of reviews. Despite the incorporation of additional analyses and methodological details, the manuscript remained clear and informative, with the same high quality of presentation, analysis, and figures. Additionally, we were pleased to hear about the plans for future work in this interesting area (e.g. further evaluation latent representations' content and their usefulness for the individual identification).Few additional comments and reflections incited by the revised manuscript and authors' replies are detailed below with hope they might be helpful for their future work.

Thank you for your thoughtful and critical insights. We greatly appreciate your recommendations and will keep them in mind for our future work.

1. Adult to neonate: how to interpret the resulting networks?On Page 5 of the response to comments file, the authors bring up an important point regarding the adult lense through which the community tends to analyse early connectomes. As mentioned by the authors, this issue is shared by the majority of papers within this space and we believe it would be useful to discuss this topic explicitly in regards to the limitations it creates for the interpretation of results. With such a point of view, do the authors envisage the latent representations derived from the infant data to be 'real' features of the functional connectome at a given time point ('in-themselves') or rather 'dissimilarities/distances' to the adult network ('in-relation-to' the training data)?

We appreciate the reviewer’s critical feedback. We believe the latent representations derived from infant rsfMRI are likely driven by both ‘real’ features of the newborn functional connectome and ‘distance’ to the adult network. Our findings suggest that the VAE could represent fundamental functional networks at a given time point or ‘real’/‘actual’ newborn networks i.e., the observance of brain networks such as the visual, auditory, and sensorimotor during the fetal-neonatal period; please also see additional findings in Figure 8 described in the response to Reviewing editor. Similarly, VAE-derived networks also reflect ‘distance’ to adult patterns, likely tracking the evolution of brain networks with maturation. Training the VAE model on a neonate only dataset will help us identify “real” vs “distance” features; this is an area of interest that we are actively investigating. In line with these, we revised our discussion as follows [page 24 line 523-534]:

“Given that the VAE model was trained to represent adult brain dynamics, it is unclear whether latent features of fetal-neonatal rsfMRI reflect “real” aspects of the fetal-neonatal functional connectome (i.e., fundamental functional networks present during different developmental periods) or are driven by similarity/dissimilarity to the adult functional connectome (i.e., the evolution of patterns of brain networks with aging). Our results suggest that both aspects are likely captured by latent features; some fundamental functional networks that remain largely consistent over neonatal period, reflecting the actual aspect of fetal-neonatal functional connectome. In contrast, some networks evolved over time (e.g., IC 6 in Figure 8); thus, some portions of their latent features may be driven by the dissimilarity/similarity to the adult network. Investigating this dichotomous aspect of VAE-derived latent representations will be another interesting topic as an application of the VAE in the fetal-neonatal neuroimaging field.”

In both of these cases, the latent representations will be practically useful, and outcompete non-linear models, with a high potential for improving the age, individual, and possibly atypical development prediction. However, the interpretation shift – VAE deriving features relevant to age prediction (supported by improved age prediction) to features relevant to brain development (suggested on Page 16 of the response to comments) – might be difficult to make until the deeper understanding of the content of derived latent representations is reached. Comparing the 30 networks derived from the models finetuned to different age ranges (as suggested by the authors) could be very interesting to shed more light on the developmental dynamism and this question. Until then, more caution might be warranted in the current setting when incorporating concepts like 'longitudinal' investigations (L81) or 'neonatal brain dynamics (L109).

Thank you for the thoughtful feedback. In the revised manuscript, we additionally compared the patterns of 30 networks in younger vs older neonates. For details, we kindly refer the reviewer to our response to Reviewing editor (RR pp. 1). We believe our additional results support the possibility of using the VAE to investigate longitudinal neurodevelopment. Nevertheless, we do agree with the reviewer that additional investigations are needed to deepen our understanding of the content of latent representations. Such studies will reinforce the findings introduced in the current work. To reflect this point, we toned down related language in the Discussion and modified the section on future directions. Please see below.

In Discussion [page 19 line 424]:

“Collectively, our proposed VAE model and analytical scheme may have the potential to serve as an important complement to existing linear computational models for disentangling complex fMRI brain patterns in fetuses and neonates for the investigation of longitudinal neurodevelopment in healthy and high-risk fetal-neonatal populations.”

And, in Discussion [page 29 line 659-664]:

“Thus, the VAE model may provide important insights into fetal-neonatal neurodevelopmental growth trajectories. In-depth assessments of ICs in the latent space, each representing a large-scale brain network, can be a very interesting research topic. Among many, one feasible assessment can be a hierarchical grouping of ICs given their pattern similarity in latent representations and compare the settled hierarchy to human brain organization. Tracking transition from one IC to other ICs over time during resting-state can be another assessment. New insights regarding the complex non-linear brain development that occurs during the prenatal-neonatal continuum may allow us to identify early biomarkers for neurobehavioral vulnerability and link specific prenatal brain developmental processes associated with aberrant neural connectivity which may underlie prevalent child and adult neurologic disorders.”

Regarding 'longitudinal' investigations (L81), we decided to remove the word “longitudinal” as below:

“Taken together, these findings suggest that investigation of fetal-neonatal FBNs using rsfMRI would likely require a computational approach that could potentially capture the non-linearity inherent to rsfMRI data into the model.”

With regards to the use “neonatal brain dynamics” (L109), we were referring to changes in brain activity rather than changes across developmental stage. To prevent possible confusion, we revised the sentence to [page 5 line 106]:

“We hypothesized that non-linear representations of fetal-neonatal rsfMRI, extracted using the VAE model pretrained with the rsfMRI of healthy adults, would carry more accurate and informative neural signatures of the prenatal-neonatal brain networks, compared to linear representations defined at the network scale using ICA or at the regional scale using multi-modal cortical parcel.”

On a slightly unrelated note, the authors then argue that 'utilizing adult rs-fcMRI at the level of the analytical model is more desirable than at the interpretation level, as there may be information from rs-fcMRI in adults that can more objectively guide the way we understand representations of neonatal rs-fcMRI'. Is this really the case? Our intuition would be that the bias introduced in one step of a pipeline is likely to be propagated in the consecutive analyses. Thus, it is difficult to see why introduction of the adult reference earlier in the research pipeline, at the level of the model, would be more desirable rather than for example on the contrary more obscuring?

First, we truly appreciate the reviewer’s critical insights. It is true that error introduced at earlier analytical step can be augmented over the chain of analysis, possibly biasing the results. However, we partially disagree with the reviewer that introducing knowledge (e.g., adult rs-fcMRI in our case) in the analytical model is not necessarily equivalent to the introduction of bias/error in the model. In fact, we believe proper incorporation of knowledge in the analytical model can suppress of error/bias in the results. One good example is a seed-based correlational analysis. The seed-based correlational analysis incorporate knowledge to the analytical model by putting a single voxel (or a brain region) as a seed region of interest. As the reviewer is already familiar, it has been widely known that the result of seed-based correlational analysis is robust against measurement error/artifact. Another counter example can be independent component analysis (ICA). While ICA is a model-free analytical tool that does not require any human knowledge, interpretation of some ICA maps can be biased by noise or artifact in data (see ^5, 6, 7^ for the vulnerability-to-noise issue of ICA). Thus, we believe our analytical strategy that incorporated knowledge from human adults (i.e., human rs-fcMRI) to the VAE model was not particularly more susceptible to noise/artifact nor yielded more errors in the results, compared to linear models. Our speculation can be supported by the superior data representation abilities of the VAE compared to linear models (Figures 2-6); yet we believe that confirmation of this aspect of the VAE awaits future study.

[Additionally, we wonder if it would be useful to have an age prediction baseline which does not use the functional inputs at all? Would comparing the age prediction performance using basic parameters (for example head size, GA at birth to predict PMA at scan) to a model trained on VAE inputs give some additional information about the VAE representational ability vs the cohort composition effects?].

This is an interesting point of view. However, we believe adding this baseline (or “ground truth” for the age prediction model) may put too much emphasis on the “age prediction” aspect of the study. To keep the study focused on our central goal – demonstrating the importance of representing non-linear features of developing fetal-neonatal networks – we decided, for the moment, to exclude additional analyses related to age prediction. We hope the reviewer concurs.

2. Radiological score and reconstruction performanceWe are hesitant regarding the presentation of the lower reconstruction performance in neonates with the radiological score=5 (Page 16 of the response to comments file, and then within the manuscript L452). The radiology score of 5 describes an 'incidental finding with possible / likely significance for both clinical and imaging analysis'. Thus, it is expected (and reassuring) that the authors observed reconstruction differences between score groups as changes to the typical brain might additionally disrupt the functional networks independently of the developmental stage of the infant, i.e. make them more unlike the adults. However, these subjects will also probably suffer from a potential decreased quality of the image processing/analysis, for example due to lesions, which could confound these results. Thus, while we think it would be very interesting to further investigate the latent representations of the infants in regards to the brain injury (for example prediction model that classifies infants into subgroups based on the radiological score within the infant group of similar age), it might be difficult to understand how to dissociate the effects of age, injury, and processing quality on reconstruction and predictive performance in this group without additional discussion and quality checks (for example does the data ans registration quality differ between different radiology score groups?). It might be more straightforward to focus on the observation of significant results in 'typical' groups (radiological score-1 and/or 2) as a support for potential of VAE to provide a surrogate measure of the distance of infant-to-adult activity pattern and clarify further the reasoning/caveats regarding the results in the score=5 group.

The reviewer presents a very strong argument against the use of images with radiology score = 5. After careful consideration, we removed statements below from our manuscript:

[page 8 Line 178-186 in the original manuscript]

“We also observed that subjects with radiological scores=5 (incidental finding with possible / likely significance for both clinical and imaging analysis; n=23; mean ± SD=0.42 ± 0.05; two-sample t-test, p=0.046 vs. group with score=1; n=172; mean ± SD=0.44 ± 0.05) showed worse reconstruction performance compared to other groups (Figure 4—figure supplementary 1).”

And [page 20 Line 452-454 in the original manuscript]

“This is supported by our finding of lower reconstruction performance in neonates with incidental MRI findings with possible / likely clinical significance (radiology score = 5) compared to other groups.”

Accordingly, the Figure 4—figure supplementary 1 was removed.

3. Foetal resultsMoreover, we remain hesitant regarding the interpretation of foetal results and therefore the use of 'foetal-neonatal' outlook throughout the manuscript.As mentioned on the L197: 'Once brain size [in foetuses] was accounted for, the negative correlation between age and reconstruction performance was reduced'. This suggests that some, but not all of this unexpected behaviour was accounted for by the head size and the ensuing ballooning effect. If we assume that the young foetuses should not be more similar to the adult cohort than the older ones in terms of the reconstruction performance, is it correct to expect additional problems remaining within the reconstruction of the foetal data?

Thank you for the feedback. Yes, there could be additional confounding factors driving the negative correlation between age and reconstruction performance in the fetal cohort. Unfortunately, our initial attempts (i.e., looking into head motion, head size, scan length as confounding factors) failed to identify the driver of this relationship. One possibility could be that the brain activity in younger fetuses is more concurrently activated/deactivated (in other words, simpler brain patterns) due to their prematurity, making the VAE reconstruct brain activity patterns better than more complex patterns observed in older fetuses. Unfortunately, we could not test this hypothesis in the current study; we, however, will keep the reviewer’s concern in mind for our future work. Regarding this matter, we revised our result as follows [page 9 line 190-195]:

“Once brain size was accounted for, the negative correlation between age and reconstruction performance was reduced (Figure 4—figure supplementary 1.E, *r*=-0.24, *p*=0.01). However, despite that the reconstruction performance was reduced, the negative correlation between fetal age and reconstruction performance remained significant. While we posit that brain representations of younger fetuses are less similar to human adults than older fetuses, our observation suggests there are additional confounding factors that we failed to identify. In contrast, brain size in newborns was not significantly associated with reconstruction performance (Figure 4—figure supplementary 1.G, *r*=0.14, *p*=0.12).”

Additionally, the statement on L40 that '[VAE led to] improved prenatal-neonatal brain maturational patterns and more accurate and stable age prediction compared to linear models' is not entirely accurate given the results presented on L280 which reports that it was cortical parcels that showed the best performance in the foetal group.

Thank you for the pointing this out. To make our statement fair, we revised our sentence as [page 2 line 40-41]:

“Here, we demonstrated that non-linear brain features, i.e., latent variables, derived with the VAE pretrained on rsfMRI of human adults, carried important individual neural signatures, leading to improved representation of prenatal-neonatal brain maturational patterns and more accurate and stable age prediction in the neonate cohort compared to linear models.”

Thus, it might be difficult to conclude on L361 that 'This finding may suggest that neurodevelopment of preterm neonates and age-matched in-utero fetuses likely differ given the early extrauterine exposure in infants born prematurely, above and beyond the difference in acquisition settings and preprocessing steps between dHCP and DBI datasets.' Although we agree with the statement in itself, because of the remaining above-mentioned issues it might be difficult to use current findings as a support.

Thank you for the critical comment. We decided to retreat our statement to pursue the rigor of our study, as concerned by the reviewer. Below is the statement removed in the revised manuscript [page 16 line 361-364 in the original manuscript].

“This finding may suggest that neurodevelopment of prematurely born neonates and age-matched in-utero fetuses likely differ given the early extra-uterine exposure in infants born preterm^8, 9^, above and beyond the difference in acquisition settings and preprocessing steps between dHCP and DBI datasets.”

Given the remaining questions regarding the reliability of the foetal results, it is possible that this population (due to very different acquisition requirements, processing problems, large developmental distance) might benefit even more than the other groups from a foetal-specific model/pipeline (whose creation was suggested by the authors). Until then, more caution might be helpful when promoting the foetal aspects of the study.

Thank you for the critical comment. Once again, we agree with the reviewer that interpretation of fetal results need extra caution. To reflect this point, we revised our discussion as below (page 22 line 489-494):

“We believe that this was unlikely, given that in our previous experiment with human adults (see supplementary Figure 1 in ^10^), we showed that the error induced by geometric reformatting was minimal (*r* ~0.98 between the original cortical pattern and reformatted/recovered cortical pattern). Lastly, unlike the neonatal cohort, we found that there was a remaining negative correlation between age and pattern reconstruction performance, which was somewhat against our expectation, even after considering head size. Thus, we believe that caution is needed in interpreting the findings with the fetal cohort. To address this issue in the fetal cohort, fetal-specific model or analytical pipeline will be needed.”

Reviewer #3 (Recommendations for the authors):The authors have provided very comprehensive responses to the comments and have largely addressed things appropriately by changing the wording in the manuscript itself.Although they do a good and compelling job showing that the VAE method is seemingly outperforming other methods (in this case ICA), I have to confess I still struggle somewhat with the insight that can be drawn from this work – apart from the "potential" of the method to study early brain development. Most prominently, the "non-linear" aspects of the method are heavily emphasised in the manuscript – which for characterising brain development would of course be a major benefit which would be unique to this method. However, as it stands, the authors concede that developmental trajectories have not been explored and it could be done in later work. Rather than suggesting there is "potential for this" however, I suspect a reader would want evidence that this method can definitely provide this kind of novel insight to conclude whether it is truly characterising biological effects and/or is worthwhile?

We appreciate the reviewer’s thoughtful feedback. As suggested by the reviewer and reviewing editor, we conducted additional analysis. Detailed summarization of results and its related methods can be found in the response to the reviewing editor. We believe that our original result/discussion and additional analysis/discussion that has been added during this revision will resolve the reviewer’s concern.

Regarding the “non-linearity” of the VAE model respect to brain development, our newly added results demonstrated that non-linearity of the VAE played an important role in performing group-wise comparison (Figure 8—figure supplement 2). When we compared the functional networks at the cortical space, brain regions spanning the whole brain, including somatosensory area, frontal area, and auditory area, were highlighted together. Unlike cortical space, however, when the comparison was conducted at the latent space level, the network difference was more focal in the primary auditory region. It is noteworthy that such discrepancy between cortical- and latent spaces was not observable in the linear models. We speculate that the distributed pattern difference in the cortical space is driven by the sample-wise variation (not representing the actual neurodevelopmental change). Change was updated:

In the result [page 18 line 395-398]:

“Functional networks of older groups showed stronger engagement of primary auditory area, with some level of lateralization toward to the right hemisphere; no change in somatosensory region was observed. When subtraction was conducted at the cortical surface (i.e., comparison between functional brain networks of two groups), older neonates showed stronger activation in somatosensory region along with engagement of the primary auditory area (Figure 8-supplementary figure 2). To summarize, our results demonstrate that the brain network engages additional brain region with advancing age (or maturity), suggesting the recharacterization of brain systems in the first weeks of the postnatal period.”

and discussion [page 29 line 654-658]:

“In contrast, the VAE model is non-linear such that combining two latent variables in the latent space can yield continuous change of network patterns in the cortical space (i.e., adding/subtracting age-related latent variable from DMN-related latent variable and projecting into the cortical space using the VAE decoder). Related to it, our findings showing a difference in age group-wise network patterns between latent vs. cortical spaces (Figure 8—figure supplement 2) suggested the possibility of the above mapping strategy that can track the neurodevelopmental growth of fetal-neonatal functional brain networks.”

This issue about demonstrating whether this method is truly a specific tool that can provide new insights about brain development are also relevant for 2 other unresolved issues – which would be key to differentiate if this is just a paper showing that a method can be used on fetal/neonatal data or if it is a method that is specifically worthwhile using on these populations. These are again are conceded by the authors but again readers I suspect, may want answered. However, I fully appreciate that these cannot be easily resolved. These issues revolve around the use of the adult training data which means: that (i) the subcortical structures cannot be included – these structures one might argue are even more important in fetuses than the cortex? and (ii) the distinction between "model generation (VAE) and interpretation (ICA)" becomes important. The authors here argue that generation is more desirable than interpretation – although I would personally argue the other way, as interpretation means that anything can be identified but can then be explored (as opposed to generation, where it would not be identified in the first place?). Age effects using ICA networks for example can be easily studied using a method like dual regression. It would be nice to see what would happen if a neonatal data set for example was used for training – but failing that I think it would benefit the reader if this specific issue is explicitly discussed in the same way as in their response?

Thank you for the critical feedback. Regarding the “subcortical structures”, we acknowledge that not including subcortical information in the VAE model is a major limitation that should be further improved/addressed in future work. We kindly refer the reviewer to our response to the editor’s comment.

Regarding the “model generation”, we want to clarify our intention in the previous response. Our intention was not to argue that model generation (VAE) is more important or useful than interpretation (ICA). Instead, we believe that the generative model may be a useful addition to existing interpretational models such as ICA, for some studies. We hope the author understands our point of view.

Regarding the “dual regression”, we agree with the reviewer that dual regression is a reasonable choice among other possible methods, for investigating group-wise difference. To clarify, the VAE model is just a one type of representation technique (such as ICA or PCA), not an analytical model itself. For this reason, after features are extracted from the VAE, conventional analytical methods such as dual regression can utilized. As demonstrated in our additional analysis shown above, dual regression was utilized to investigate the group-wise difference (for details, we kindly refer the reviewer to our above response to reviewing editor’s comment).

Lastly, regarding the limitation of the VAE model and future direction of the VAE model, we believe we thoroughly discussed and addressed this during the last round of revision. While we respectfully acknowledge the reviewer’s stance, we believe the current format of our Discussion section (limitations followed by future work) is the optimal structure to deliver our message. We hope the reviewer is amenable to our point of view.

**References**

1. Fair DA*, et al.* Functional brain networks develop from a “local to distributed” organization. *PLoS computational biology* 5, e1000381 (2009).

2. Dosenbach NU*, et al.* Prediction of individual brain maturity using fMRI. *Science* 329, 1358-1361 (2010).

3. Fransson P*, et al.* Spontaneous brain activity in the newborn brain during natural sleep—an fMRI study in infants born at full term. *Pediatric research* 66, 301-305 (2009).

4. Lordier L*, et al.* Music processing in preterm and full-term newborns: a psychophysiological interaction (PPI) approach in neonatal fMRI. *Neuroimage* 185, 857-864 (2019).

5. Griffanti L*, et al.* Hand classification of fMRI ICA noise components. *Neuroimage* 154, 188-205 (2017).

6. Salimi-Khorshidi G, Douaud G, Beckmann CF, Glasser MF, Griffanti L, Smith SM. Automatic denoising of functional MRI data: combining independent component analysis and hierarchical fusion of classifiers. *Neuroimage* 90, 449-468 (2014).

7. Griffanti L*, et al.* ICA-based artefact removal and accelerated fMRI acquisition for improved resting state network imaging. *Neuroimage* 95, 232-247 (2014).

8. De Asis-Cruz J*, et al.* Functional brain connectivity in ex utero premature infants compared to in utero fetuses. *NeuroImage* 219, 117043 (2020).

9. Bouyssi-Kobar M*, et al.* Third trimester brain growth in preterm infants compared with in utero healthy fetuses. *Pediatrics* 138, (2016).

10. Kim J-H, Zhang Y, Han K, Wen Z, Choi M, Liu Z. Representation learning of resting state fMRI with variational autoencoder. *NeuroImage* 241, 118423 (2021).